



**Using airborne observations to improve estimates of short-lived halocarbon emissions**
**during summer from Southern Ocean**
Elizabeth Asher[1], Rebecca S. Hornbrook[1], Britton B. Stephens[1], Doug Kinnison[1], Eric J. Morgan[5], Ralph F.
Keeling[5], Elliot L. Atlas[6], Sue M. Schauffler[1], Simone Tilmes[1], Eric A. Kort[2], Martin S. Hoecker-Martínez[3],
Matt C. Long[1], Jean-François Lamarque[1], Alfonso Saiz-Lopez[4,1], Kathryn McKain[7,8], Colm Sweeney[8], Alan J.
Hills[1], and Eric C. Apel[1]
[1] National Center for Atmospheric Research, Boulder, Colorado, USA
[2] University of Michigan, Climate and Space Sciences and Engineering, Ann Arbor, Michigan, USA
[3] University of Redlands, Physics Department, Redlands, California, USA
[4] Department of Atmospheric Chemistry and Climate, Institute of Physical Chemistry Rocasolano, CSIC,
Madrid, Spain
[5] Scripps Institution of Oceanography, University of California, San Diego, California, USA
[6] University of Miami, Department of Atmospheric Sciences, Miami, Florida, USA
[7] Cooperative Institute for Research in Environmental Sciences, University of Colorado, Boulder,
Colorado, USA
[8] National Oceanic and Atmospheric Administration, Boulder, Colorado, USA



**Abstract.**
We present observations of $CHBr_3$, $CH_2Br_2$, $CH_3I$, $CHClBr_2$, and $CHBrCl_2$ from the Trace Gas
Organic Analyzer (TOGA) during the $O_2/N_2$ Ratio and $CO_2$ Airborne Southern Ocean (ORCAS)
study and the 2nd Atmospheric Tomography mission (ATom-2), in January and February of 2016
and 2017. We also use $CH_3Br$ from the University of Miami Advanced Whole Air Sampler
(AWAS) on ORCAS and from the UC Irvine Whole Air Sampler (WAS) on ATom-2. We
compare our observations with simulations from the Community Atmosphere Model with
Chemistry (CAM-Chem). We report regional enrichment ratios of $CHBr_3$ and $CH_2Br_2$ to $O_2$ of
$0.19 \pm 0.01$, and $0.07 \pm 0.004$ pmol : mol, poleward of 60° S between 180° W and 55° W, and of
$0.32 \pm 0.02$, $0.07 \pm 0.004$ pmol : mol over the Patagonian Shelf, between 40° S and 55° S and
between 70° W and 55° W where we also report enrichment ratios of $CH_3I$ to $O_2$ of $0.38 \pm 0.03$
pmol : mol and of $CH_2ClBr_2$ to $O_2$ of $0.19 \pm 0.04$ pmol: mol. Using the Stochastic Time-Inverted
Lagrangian Transport (STILT) particle dispersion model, we use correlations between
halogenated hydrocarbon mixing ratios and the upwind influences of chlorophyll $a$, sea ice, solar
radiation, and dissolved organic material to investigate previously hypothesized sources of
halogenated volatile organic compounds (HVOCs) in the southern high latitudes. Our results are
consistent with a biogenic regional source of $CHBr_3$, and both non-biological and biological
sources of $CH_3I$ over these regions, but do not corroborate a regional sea-ice source of HVOCs in
January and February. Based on these relationships, we estimate the average two-month (Jan.-
Feb.) emissions poleward of 60° S between 180° W and 55° W of $CHBr_3$, $CH_2Br_2$, $CH_3I$, and
$CHClBr_2$ to be $91 \pm 8$, $31 \pm 17$, $35 \pm 29$, and $11 \pm 4$ pmol m$^{-2}$ hr$^{-1}$, and regional emissions of these
gases over the Patagonian Shelf to be $329 \pm 23$, $69 \pm 5$, $392 \pm 32$, $24 \pm 4$ pmol m$^{-2}$ hr$^{-1}$
respectively.

**1 Introduction**
Emissions of halogenated volatile organic compounds (HVOCs) influence regional atmospheric
chemistry and global climate. Through the production of reactive halogen radicals at high
latitudes, HVOCs contribute to tropospheric and stratospheric ozone destruction, and alter the
sulfur, mercury, nitrogen oxide and hydrogen oxide cycles (e.g. WMO, 2011; von Glasow and
Crutzen; 2007, Saiz-Lopez et al., 2007; Bloss et al., 2005; Boucher et al., 2003; Schroeder et al.,
1998; Obrist et al., 2011). Indeed, HVOCs may be among the most important sources of
inorganic bromine to the whole atmosphere, since recent evidence indicates that sea salt is scarce
and insufficient to affect the bromine budget in the middle and upper troposphere (Murphy et al.,
in review).
Phytoplankton and macroalgae in the ocean are the main sources to the atmosphere of several
very short-lived bromocarbons, including bromoform ($CHBr_3$), dibromomethane ($CH_2Br_2$),
dibromochloromethane ($CHClBr_2$), and bromodichloromethane ($CHBrCl_2$) (Moore et al., 1996;
Carpenter et al. 2003; Butler et al., 2007; Raimund et al., 2011). Other HVOCs, such as methyl
iodide ($CH_3I$), and methyl bromide ($CH_3Br$) have many natural sources, such as coastal
macroalgae, phytoplankton, the temperate forest soil and litter, and biomass burning (e.g., Bell et



al., 2002; Sive et al., 2007; Colomb et al. 2008; Drewer et al., 2008).  CH$_3$I is also formed through non-biological reactions in surface seawater, and CH$_3$Br is emitted as a result of anthropogenic crop fumigation (e.g., Moore and Zafiriou; 1994, WMO 2014).  Over the Southern Ocean specifically, hypothesized sources of HVOCs include: coastal macroalgae, phytoplankton, sea ice algae, and photochemical or dust stimulated non-biological production at the sea surface (e.g., Manley and Dastoor 1998; Moore and Zafirou 1994; Richter and Wallace 2004; Williams et al., 2007; Moore et al., 1996; Tokarczyk and Moore 1994; Sturges et al., 1992).

We owe our current understanding of marine HVOCs at high latitudes in the Southern Hemisphere largely to ship-based field campaigns and laboratory process studies (e.g., Abrahamsson et al. 2004a,b; Atkinson et al., 2012; Carpenter et al., 2007; Moore et al., 1996; Chuck et al., 2005; Butler et al., 2007; Raimund et al., 2011; Hughes et al., 2009; Hughes et al., 2013).  These studies have demonstrated that the marine boundary layer (MBL) contains elevated levels of several HVOCs, and that numerous biological and non-biological sources of HVOCs exist.  These studies indicate moderate ocean sources of CHBr$_3$ and CH$_2$Br$_2$ at high latitudes in the Southern Hemisphere, which are often underestimated in global atmospheric models (Hossaini et al., 2013; Ordoñez et al., 2012; Ziska et al., 2013).  Ship-based and Lagrangian float observations provide invaluable information on the sources and temporal variability of compounds in the surface ocean.  These methods offer the advantage of simultaneous measurements of both air and seawater to evaluate the gases' saturation state in the surface ocean.  Yet ship-based measurements onboard these slow moving platforms also have drawbacks: they under sample the spatial variability of HVOCs (e.g., Butler et al., 2007) and require assumptions about gas-exchange rates to estimate fluxes.

Large-scale atmospheric observations of HVOCs are needed to understand the influence of atmospheric transport as well as the spatial variability of ocean sources on their distributions. At low latitudes, large-scale convection at the intertropical convergence zone carries bromocarbons and other HVOCs into the free troposphere and lower stratosphere (e.g., Liang et al., 2014; Navarro et al., 2015).  In polar regions, however, vertical transport is more limited in scale. Small, convective plumes may form over the marginal sea ice zone, related to sea ice leads as well as winds from ice to open-waters (e.g. Schnell et al. 1989). To a large extent, however, polar regions are characterized by stable boundary layers in summer. Although vertical transport within and across a stable boundary layer remains poorly understood, wind shear, internal gravity waves, and frontal systems create turbulence that contributes to vertical mixing (e.g. Anderson et al. 2008). Given their extended photochemical lifetimes at high latitudes (see Sect. 2.5 for a brief discussion), zonal transport as well as vertical transport could have a large impact on vertical gradients of HVOCs.

Aircraft observations can rapidly map basin-wide vertical distributions, support quantitative air-sea flux estimates, and provide spatial constraints to atmospheric models, but rarely address the temporal variability in mixing ratios or emissions.  Few constraints on HVOC mixing ratios or emissions based on airborne data exist at high latitudes in the Southern Hemisphere.  Two earlier aircraft campaigns that have measured summertime HVOCs in this region are the first Aerosol Characterization Experiment (ACE-1; Bates et al., 1999) and the first High-performance





Instrumented Airborne Platform for Environmental Research (HIAPER) Pole-to-Pole
Observations (HIPPO; Wofsy, 2011) campaign. For these two aircraft campaigns, whole air
samples were collected onboard the NSF/NCAR C-130 and the NSF/NCAR Gulfstream V (GV)
during latitudinal transects over the Pacific Ocean as far south as 60° S and 67° S, respectively.
However, the ACE-1 and HIPPO campaigns obtained relatively few whole air samples in this
region, with ≤100 samples poleward of 60° S combined (e.g., Blake et al., 1999; Hossaini et al.,
2013). ACE-1 measurements of $CH_3I$ in the MBL indicate a strong ocean source between 40° S
and 50° S in austral summer, with mixing ratios above 1.2 nmol mol$^{-1}$ below ~1 km (Blake et al.,
109  1999).

HVOCs are frequently incorporated into earth system climate models, using either climatologies
or parameterizations based on satellite observations of chlorophyll and geographical region. This
study uses airborne observations, geophysical datasets, and a Lagrangian atmospheric transport
model to investigate HVOC distributions at high southern latitudes, evaluate existing
parameterizations of HVOC emissions in a global atmospheric chemistry transport model, assess
contributions from previously hypothesized regional sources for the Southern Ocean, and
provide new means of estimating HVOC emissions based on relationships between airborne
observations and modeled or remotely sensed surface parameters.

**2 Observations**
**2.1 Overview**
Atmospheric measurements for this study were collected at high latitudes in the Southern
Hemisphere as part of the $O_2/N_2$ Ratio and $CO_2$ Airborne Southern Ocean (ORCAS) study
(Stephens et al., 2018), and the second NASA Atmospheric Tomography Mission (ATom-2),
near Punta Arenas, Chile (Fig. 1).  The ORCAS field campaign took place from Jan. 15 – Feb.
29, 2016 onboard the NSF/NCAR GV.  On Feb. 10 and 13, 2017 the NASA DC-8 aircraft passed
over the eastern Pacific sector poleward of 60° S (defined here as Region 1) on the sixth research
flight and over the Patagonian Shelf between 40° S and 55° S and between 70° W and 55° W
(defined here as Region 2) on the seventh research flight of ATom-2 from Christchurch, New
Zealand to Punta Arenas and from Punta Arenas to Ascension Island, respectively.  The two
regions for this study are defined based loosely on dynamic biogeochemical provinces identified
using bathymetry, algal biomass, sea surface temperature and salinity (Reygondeau et al. 2013).
Both projects featured en route vertical profiling from near the ocean surface (~ 150 m) to the
upper-troposphere, with 74 ORCAS and seven ATom-2 (during the sixth and seventh flights)
low-attitude level legs in the MBL. These campaigns shared a number of instruments, including
the NCAR Trace Gas Organic Analyzer (TOGA), the NCAR Atmospheric Oxygen (AO2)
instrument, a Picarro cavity ringdown spectrometer operated by NOAA, discussed below. More
information about individual instruments may be found in Stephens et al., 2018 and at
https://www.eol.ucar.edu/field_projects/orcas and https://espo.nasa.gov/atom/content/ATom.





### 2.2 Halogenated VOCs

During ORCAS and ATom-2 TOGA provided mixing ratios of over 60 organic compounds, including HVOCs, at background levels. The instrument, described in Apel et al. (2015), continuously collects and analyzes samples with a 35-second sampling period and repeats the cycle every two-minutes using online fast gas chromatography and mass spectrometry. HVOCs reported here have an overall ±15% relative accuracy and ±3% relative precision, and detection limits of $\leq$ 0.2 ppt for $CHBr_3$, $CH_2Br_2$, $CHClBr_2$, $CHBrCl_2$, and $CH_3I$. This study also leverages measurements of $CH_3Br$ with a detection limit of 0.2 ppt from whole air samples from the U. Miami / NCAR Advanced Whole Air Sampler (AWAS; Schauffler et al., 1999) onboard the GV during the ORCAS campaign and the UC Irvine Whole Air Sampler (WAS; Blake et al., 2001) onboard the DC-8 during the ATom-2 campaign. In addition, comparisons between onboard collected whole air samples and in-flight TOGA measurements, when sharing over half of their sampling period with TOGA measurements, showed good correlations for $CHBr_3$, $CH_2Br_2$, $CH_3I$, and $CHClBr_2$, although there were some calibration differences (Fig. S1 and Fig. S2). In addition to the comparison between co-located atmospheric measurements, we also conducted a lab inter-comparison following the campaign between NOAA's programmable flask package (PFP) and TOGA (Table S1; see supplement for details).

### 2.3 $\delta(O_2/N_2)$ and $CO_2$

The AO2 instrument measures variations in atmospheric $O_2$, which are reported as relative deviations in the oxygen to nitrogen ratio ($\delta(O_2/N_2)$), following a dilution correction for $CO_2$ (Keeling et al., 1998; Stephens et al. 2018). The instrument's precision is ± 2 per meg units (one in one million relative) for a 5 second measurement (Stephens et al., 2003; Stephens et al., manuscript in preparation, 2019). Anthropogenic, biogenic, and oceanic processes introduce $O_2$ perturbations that are superimposed on the background concentrations of $O_2$ in air ($XO_2$, in dry air = 0.2093). $O_2$ is consumed when fossil fuels are burned and produced during terrestrial photosynthesis. Seasonal changes in the ocean heat content lead to small changes in atmospheric $N_2$. As others have done (Keeling et al., 1998; Garcia and Keeling, 2001; Stephens et al., 2018), we isolated the air-sea $O_2$ signal by subtracting model estimates of the terrestrial photosynthesis, fossil-fuel combustion, and air-sea $N_2$ flux influences from the $\delta(O_2/N_2)$ measurement (Equation 1). The difference of the $\delta(O_2/N_2)$ measurement and these modeled values is multiplied by $XO_2$ to convert to ppm equivalents as needed (ppm eq; Keeling et al., 1998; Equation 1).

$$O_{2\text{-ppm-equiv}} = [\delta(O_2/N_2) - \delta(O_2/N_2)_{Land} - \delta(O_2/N_2)_{Fossil\ Fuel} - \delta(O_2/N_2)_{N2}] \times XO_2 \qquad (1)$$

We obtained the modeled $\delta(O_2/N_2)$ signal land influences from the land model component of the Community Earth System (CESM), the fossil fuel combustion influences from the Carbon Dioxide Information Analysis Center (CDIAC; Boden et al. 2017), and the air-sea $N_2$ influences from the oceanic component of CESM. These fluxes were all advected through the specified dynamics version of CAM, as described below in Sect. 3.1 and in Stephens et al. (2018). The $XO_2$ in 2016 is the Tohjima et al. (2005) value from the year 2000 adjusted for the 4 ppm $yr^{-1}$ or ~20 per meg $yr^{-1}$ decrease in $O_2$ between 2000 and 2016.




$CO_2$ measurements were provided by NOAA's Picarro G2401-m cavity ring down spectrometer
modified to have a ~1.2 sec measurement interval and a lower cell pressure of 80 Torr, which
enabled the instrument to function at the full range of GV altitudes. (McKain et al., in prep.,
2019). Dry-air mole fractions were calculated using empirical corrections to account for dilution
and broadening effects in the laboratory before and after the campaign deployments, and in-flight
calibrations were used to determine an offset correction for each flight.  Corrected $CO_2$ data have
a total average uncertainty of 0.07 ppm (McKain et al. in prep., 2019).  To merge them with the
TOGA data, these faster $O_2$ and $CO_2$ measurements were arithmetically averaged over TOGA's
35-s sampling periods (Stephens et al., 2017 and https://espo.nasa.gov/atom/content/ATom).

**2.4 Observed HVOC patterns and relationships**
Zonal cross-sections of HVOC data collected on ORCAS and ATom-2 illustrate unprecedented
spatial sampling across our study area between the surface and 12 km (Fig. 2). Above average
mixing ratios of $CH_3I$, $CHBr_3$, and $CHClBr_2$, typically remain confined to the lower ~2-4 km of
the atmosphere (Fig. 2a, b, and d). These compounds have lifetimes of approximately two
months or less.  Conversely, weak sources and longer lifetimes (≥ 3 months) may have
contributed to similar concentrations of $CH_2Br_2$ and $CHBrCl_2$ throughout the troposphere and
above average mixing ratios as high as 8 km (Fig. 2c, e). Unfortunately, the availability of data
above the detection limit and absence of BL enhancements for $CHBrCl_2$ preclude the
identification of strong regional sources at this time. Meridional distributions also indicate lower
latitude sources of $CH_3I$ and $CH_3Br$ (≥ -50°), potentially resulting from terrestrial and
anthropogenic contributions, and higher latitude sources (≤ -60°) of $CHBr_3$, $CH_2Br_2$, and
$CHClBr_2$ (Fig. 2a-d,f).
Across our study area in both 2016 and 2017, we found that $CHBr_3$ and $CH_2Br_2$ exhibit a
consistent enhancement ratio with each other in the MBL both in Region 1 and Region 2 (Fig.
3a, c), which suggests that these species may be co-emitted. Previous studies have documented
co-located source regions of $CHBr_3$ and $CH_2Br_2$ in the Southern Ocean (e.g. Hughes et al., 2009;
Sturges et al., 1993), and laboratory studies indicate that phytoplankton and their associated
bacteria, including a diatom species isolated from coastal waters along the Antarctic Peninsula
and common to the Southern Ocean, are capable of emitting both $CHBr_3$ and $CH_2Br_2$ (Hughes et
al., 2013; Tokarczyk and Moore 1994).  We note that the non-linearity observed in ratios of these
two gases at low $CHBr_3$ levels likely reflects the differences in emissions during strong
phytoplankton blooms, as oppose to other periods.  For instance, Huges et al. (2013) also report
distinct seawater slopes between $CH_2Br_2$ to $CHBr_3$, when chl *a* was increasing. Mixing ratios of
$CHBr_3$ and $CHClBr_2$ were also correlated (Fig. 3d) in Region 2, and, a similar, weaker
relationship was observed in Region 1 (Fig. 3b).  $CHClBr_2$ is a less well-studied compound than
$CH_2Br_2$.  Yet these consistent relationships suggest that $CHBr_3$ and $CHClBr_2$ may either share
some of the same sources or have sources that co-vary.
**2.5 Observed HVOC relationships to δ($O_2$/$N_2$) and $CO_2$**



For these comparisons, both $O_2$ and $CO_2$ mixing ratios from the upper troposphere (5-7 km) were
subtracted from the data to detrend for seasonal and inter-annual variability (Fig. 4; Fig. S4). In
Fig. 4 we present type II major axis regression fits to data between the ocean surface and the
lowest 7 km for bromocarbons with photochemical lifetimes of ≥ 1 month and from the lowest 2
km for $CH_3I$ with a photochemical lifetime of ∼ 1 week. We used a type II major axis regression
model (bivariate) to balance the influence of measurement uncertainty in HVOCs (on the y-axis)
and the measurement uncertainty in $O_2$ and $CO_2$ (on the x-axis) on the regression slope (Ayers et
al. 2001; Glover et al., 2011). As noted by previous studies, simple least squares linear
regressions fail to account for uncertainties in predictor variables (e.g. Cantrell et al. 2008).
The robust correlations of $CHBr_3$ and $CH_2Br_2$ with $\delta(O_2/N_2)$, in both 2016 and 2017 and in
Region 1 and Region 2, provides support for a regional biogenic source of these two HVOCs
(Fig. 4a, b and Fig. 4d, e). The air-sea exchange of oxygen during summer in the Southern
Ocean is driven by net community production (the excess of photosynthesis over respiration) in
the surface mixed layer, surface warming, and to a lesser extent ocean advection and mixing (e.g.
Stephens et al., 1998; Tortell and Long 2009; Tortell et al., 2014). Note that we adjust for
influences on the $\delta(O_2/N_2)$ from thermal $N_2$ fluxes (see Equation 1, Sect. 2.3 for details).
Biological $O_2$ supersaturation in the surface mixed layer develops quickly in the first several
days of a phytoplankton bloom and diminishes as community respiration increases and air-sea
gas exchange equilibrates the surface layer with the atmosphere on a timescale of ∼ 1 week.
$CHBr_3$ (and $CH_2Br_2$) is emitted from phytoplankton during the exponential growth phase
(Hughes et al., 2013), which often coincides with high net community production and the
accumulation of $O_2$ in surface waters. The bulk air-sea equilibration time for an excess of $CHBr_3$
and other HVOCs is also similar to $O_2$, although the photochemical loss of HOVCs will alter
their ratio over time.
Our observations suggest a biological source for $CHBr_3$ and $CH_2Br_2$ in Region 1 (Fig. 4a and Fig.
4b). In contrast to $CHBr_3$ and $CH_2Br_2$, we observe a weaker relationship between $CH_3I$ and $O_2$ in
Region 1 (Fig. 4c), consistent with the existence of other, non-biological sources of $CH_3I$ in this
region. Figure 4d-f illustrates strong relationships between all three HVOCs and $O_2$ in Region 2.
This implies that the dominant source of $CH_3I$ emissions over the Patagonian shelf is biological.
The slope of the regression between $CHBr_3$ and $O_2$ also changes noticeably between Region 1
and Region 2. Molar enrichment ratios are 0.20 ± 0.01, and 0.07 ± 0.004 nmol : mol for $CHBr_3$
and $CH_2Br_2$ to $O_2$ in Region 1, and 0.32 ± 0.02, 0.07 ± 0.004 pmol : mol in Region 2. In Region
2, we also report enrichment ratios of $CH_3I$ to $O_2$ of 0.38 ± 0.03 pmol : mol, based on the
correlation in Figure 4f.
In contrast to $O_2$, air-sea fluxes of $CO_2$ over the Southern Ocean during summer reflect the
balance of opposing thermal and biological drivers (e.g. Stephens et al., 1998; 2018). Ocean
buffering chemistry results in $CO_2$ equilibration across the air-sea interface on a timescale of
several months. ORCAS observations showed a depletion of $CO_2$ in the MBL, indicating that
uptake driven by net photosynthesis dominated over thermally driven outgassing during the
several months preceding the campaign (Stephens et al. 2018). $CHBr_3$ and $CH_2Br_2$ in the lowest
7 km were negatively correlated with $CO_2$ in both years in Region 1 and Region 2 (Fig. S3a, b, d,




e). Interestingly, $CH_3I$ was not correlated with $CO_2$ in Region 1, likely due to the long air-sea
equilibration timescale of $CO_2$ compared with a 9-day air-sea equilibration time and a 7-day
photochemical lifetime for $CH_3I$ (see Supplement for details on calculations of bulk sea air
equilibration times). For longer lived species, correlations for HVOCs to $CO_2$ have similar $r^2$-
values as those for HVOCs to $\delta(O_2/N_2)$, but model and climatological estimates of Southern
Ocean $CO_2$ fluxes are much less certain than for $O_2$ (Anav et al., 2015; Nevison et al., 2016). As
a result, we use modeled $O_2$ fluxes as the basis for our HVOC flux estimates (see Sect. 5.1 for
details).
**3 CAM-Chem Evaluation**
**3.1 CAM-Chem Model Configuration**
The Community Earth System Model version 1 (CESM1), Community Atmosphere Model with
chemistry (CAM-Chem) is a global three-dimensional chemistry climate model that extends
from the Earth's surface to the stratopause. CAM-Chem version 1.2 includes all the physical
parameterizations of CAM4 (Neale et al., 2013) and a finite volume dynamical core (Lin, 2004)
for tracer advection. The model has a horizontal resolution of 0.9° latitude × 1.25° longitude,
with 56 vertical hybrid levels and a time-step of 30 minutes. Meteorology is specified using the
NASA Global Modeling and Assimilation Office (GMAO) Goddard Earth Observing System
Model, Version 5 (GEOS-5; Rienecker et al., 2008) (GEOS-5), following the specified dynamic
procedure described by Lamarque et al. (2012). Winds, temperatures, surface pressure, surface
stress, and latent and sensible heat fluxes are nudged using a 5-hour relaxation timescale to
GEOS-5 1° × 1° meteorology. The sea surface temperature boundary condition is derived from
the Merged Hadley-NOAA Optimal Interpolation Sea Surface Temperature and Sea-Ice
Concentration product (Hurrell et al., 2008). The model uses chemistry described by Tilmes et
al. (2016), biomass burning and biogenic emissions from the FINN and MEGAN 2.1 products
(Guenther et al., 2012) with additional tropospheric halogen chemistry described in Fernandez et
al. (2014) and Saiz-Lopez et al. (2014), including ocean emissions of $CHBr_3$, $CH_2Br_2$, $CHBr_2Cl$,
and $CHBrCl_2$, with parameterized emissions based on chlorophyll $a$ (chl $a$) concentrations and
scaled by a factor of 2.5 over coastal regions, as opposed to open ocean regions. The model used
an existing $CH_3I$ flux climatology (Bell et al., 2002), and $CH_3Br$ was constrained to a surface
lower boundary condition, also described by Ordoñez et al., (2012). This version of the model
was run for the period of the ORCAS field campaign (January and February 2016), following a
24-month spin-up. To facilitate comparisons to ORCAS observations, output included vertical
profiles of modeled constituents from the two nearest latitude and two nearest longitude model
grid-points (four profiles in total) to the airborne observations at every 30-min model time-step.
Following the run, simulated constituent distributions were linearly interpolated to the altitude,
latitude and longitude along the flight track, yielding co-located modeled constituents and
airborne observations. This version of the model has not yet been run for the ATom-2 period.

**3.2 Model-Observation comparisons**





The ORCAS dataset provides an exceptional opportunity to evaluate the CAM-Chem HVOC
emission scheme (Ordoñez et al., 2012) at high latitudes in the Southern Hemisphere.  We
compared modeled HVOC constituents to corresponding observations along the ORCAS flight
track (Fig. 5; Fig. 6).   In these figures, we used type II major axis regression models to balance
the measurement uncertainty (on the y-axis) and the inherent, yet difficult to quantify
representativeness and errors in a global atmospheric chemistry model (on the x-axis). We note
that this comparison may favor constituents with longer photochemical lifetimes, when transport
and mixing dominate over source heterogeneity.
In Region 1 and Region 2, both the model and observations indicate that elevated mixing ratios
of $CH_3I$ remain confined to the MBL (Fig. 5a and Fig. 6a), presumably due to its relatively short
photochemical lifetime.  Modeled and observed $CH_3I$ are poorly correlated in Region 1 ($r^2$ =
0.20; Fig. 5b) and better correlated in Region 2 ($r^2$ = 0.70; Fig. 6b).  In both regions, the model
under predicts $CH_3I$ above the MBL, which may indicate slower observed photochemical loss
than the model predicts. We found strong correlations and agreement to within a factor of ~2
between modeled and observed $CHBr_3$ and $CH_2Br_2$ (Fig. 5c -f and Fig. 6c-f). Relatively long
lifetimes (≥ 1 month) in Region 1 likely enable vertical and zonal transport of $CHBr_3$ and $CH_2Br_2$
to the mid and upper troposphere (Fig. 5c and e).  The model was biased low with respect to
measurements of $CH_3Br$ by ~25% in Region 1 and Region 2 (Fig. 5g-h and Fig. 6g-h),
potentially as a result of an incorrect surface lower boundary condition.  The model
underpredicted the mean vertical gradient in $CHClBr_2$, although it did a reasonable job of
representing the mean vertical gradient in $CHBrCl_2$, in both Region 1 and Region 2.  In both
cases, however, the model failed to capture the spatial variability in both $CHClBr_2$ and $CHBrCl_2$
observations (Fig. 5i-l and Fig. 6i-l).  Region 2 contains stronger sources of HVOCs than Region
1, which has been documented in numerous ship-based campaigns and archived in the
Halocarbons in the Ocean and Atmosphere database (HalOcAt; https://halocat.geomar.de/).
Region 2 also has much higher chl $a$ (Fig. S4), supporting biogenic sources for these gases.

**4 Geophysical Surface Influences**
**4.1 STILT model Configuration**
The Stochastic Time-Inverted Lagrangian Transport (STILT; Lin et al., 2003) particle dispersion
model uses a receptor oriented framework to infer surface sources or sinks of trace gases from
atmospheric observations collected downstream, thus simulating the upstream influences that are
ultimately measured at the receptor site. The model tracks ensembles of particle trajectories
backward in time and the resulting distributions of these particles can be used to define surface
influence maps for each observation. STILT was run using 0.5º GDAS reanalysis winds to
investigate the transport history of air sampled along the flight track (Stephens et al., 2018).  For
each TOGA observation, an ensemble of 4,096 particles was released from the sampling location
and followed over a backwards simulation period of seven days.  Particles in the lower half of
the simulated MBL are assigned a surface influence value, which quantitatively links observed
mixing ratios to surface sources (Lin et al., 2003).  The average surface influence of all 4,096



particles per sampling location yields an hourly and spatially gridded surface influence functions
(ppt m$^2$ s pmol$^{-1}$) at a spatial resolution of 0.25° x 0.25° for each sample point.
Uncertainty in the surface influence functions is strongly influenced by the accuracy of the
underlying meteorological transport. We evaluated the GDAS reanalysis winds by comparing
model winds interpolated in space and averaged between corresponding time points and pressure
levels to match aircraft observations. By evaluating observed winds compared with modeled
winds along the flight tracks we can estimate uncertainty in the surface influence functions. We
consider the observation-model differences in both wind speed and direction to approximate
errors in surface influence strength and location. For wind speed, a small bias may be present,
where we find a median difference between observations and reanalysis of 0.68 m/s, a 5%
relative bias. The 1-sigma of the wind speed difference is 2.3 m/s, corresponding to a 19% 1-
sigma uncertainty in wind speed. In its simplest approximation, the wind speed error will
correlate with surface influence error, and thus we take 19% as an approximation of the surface
influence strength uncertainty. We consider the wind direction error to evaluate the possible size
of spatial errors in footprint location. We find a 1-sigma error of 14 degrees in wind speed.
Given median wind speeds in this domain, this corresponds to a possible error of 260 km/day
possible error.

### 4.2 Ancillary Data

For this study, remotely sensed and reanalysis data were used with STILT influence functions in
linear and multi-linear regressions to explain observed mixing ratios of CHBr$_3$, CH$_2$Br$_2$, CH$_3$Br
and CH$_3$I. These data included a combination of chl $a$, sea ice concentration, absorption due to
ocean detrital material, and downward shortwave radiation at the ocean surface.
We used daily sea ice concentration data (https://nsidc.org/data/nsidc-0081) at a 25 km x 25 km
spatial resolution between 39.23° S and 90° S, 180° W – 180° E from the NASA National Snow
and Ice Data Center Distributed Active Archive Center (NSIDC; Maslanik et al., 1999). This
data reports the fraction of sea-ice cover, land-ice cover, and open water. Unfortunately, these
data do not provide any information on sea ice thickness, or the presence of brine channels or
melt ponds, which may modulate emissions from sea-ice covered regions. Sea ice concentration
data were calculated using measurements of near-real-time passive microwave brightness
temperature from the Special Sensor Microwave Image/Sounder (SSMIS) on the Defense
Meteorological Satellite Program (DMSP) satellites. NSIDC sea ice concentration data were
arithmetically averaged to yield 0.25° x 0.25° binned sea ice fraction for use with gridded surface
influence functions.
Due to persistent cloud cover over the Southern Ocean, which often precludes the retrieval of
remotely sensed ocean color data, we used 8-day mean composite Aqua MODIS L3 distributions
of chl $a$ (OCI algorithm) and absorption due to gelbstoff and detrital material at 443 nm and its
uncertainty (GIOP model; NASA Goddard Space Flight Center, 2014). Absorption due to
gelbstoff and detrital material at 443 nm is used as a proxy for colored dissolved organic matter
(CDOM; https://oceancolor.gsfc.nasa.gov/atbd/giop/). CDOM is hypothesized to be an important
source of carbon for the photochemical production of CH$_3$I (Moore et al., 1994). Raw 4 km x 4



km data were geometrically averaged, based on lognormal probability density functions, to a
spatial resolution of 0.25° x 0.25° for use with gridded surface influence functions.  We used the
ratio of the 0.25° x 0.25° gridded uncertainty in the detrital material absorption to the absorption
as the relative uncertainty for flux calculations (see Sect. 5.2).
The National Center for Environmental Prediction (NCEP) provides Final Global Data
Assimilation System (GDAS/FNL) global data of downward shortwave radiation at the surface
at 0.25 degree and 6-hour resolution  (NCEP, 2015).  We chose downward shortwave radiation
for use with gridded surface influence functions because the photo-production of $CH_3I$ has been
observed at all visible wavelengths (Moore et al., 1994).  This reanalysis data is available at a
higher temporal resolution and better spatial coverage than satellite retrievals of PAR or
temperature.

### 4.3 Relationships between predicted influences and observations

We used STILT to explore the relationships between observed mixing ratios and the upstream
geophysical influence functions (Equations 2-3) of sea ice, chl $a$, absorption due to detritus, and
downward shortwave radiation at the surface, which relate to various regional hypothesized
sources of HVOCs such as marine phytoplankton, phytoplankton in sea ice brines, and
decomposing organic matter in surface seawater (e.g. Moore and Zafirou 1994; Moore et al.,
1996; Tokarczyk and Moore 1994; Sturges et al., 1992).  These relationships can help evaluate
the likelihood of particular HVOC sources, and in the case of statistically significant correlations
may be used to derive an estimated flux field (See Sect. 5.2 for details).

We tested whether observed mixing ratios (Z) could be explained by a linear relationship in
which the predictor variable is the product of the surface influence function (H) and a potential
geophysical source distribution (s), such as chl $a$, as well as an intercept (b), a slope (a), and error
term $\xi$ (Equation 2; Fig S5). Moreover, this relationship can be generalized as a multiple linear
regression with multiple surface influence functions ($HS_1$, $HS_2$…) and slope coefficients ($a_1$, $a_2$;
Equation 3), when HVOC mixing ratios may be related to multiple gridded geophysical
influence functions (Hs).  The multiple linear regression may also include an interaction term
($HS_1HS_2$) between predictor variables (e.g. $HS_1$ and $HS_2$) with a slope coefficient ($a_3$) to improve
the fit.  Statistical correlations between mixing ratios and geophysical influence functions are
used to support or reject hypothesized sources.  A flux ($\mu mol\ m^{-2}\ s^{-1}$) may be then estimated for
each grid cell based on the product of the slopes ($a_1$, $a_2$…) and the potential source distributions
($Hs_1$, $Hs_2$…).  Grid cell fluxes are averaged over a geographical region to yield the average
regional flux.  We used the standard deviation of the regression coefficients and the relative
uncertainty in the source fields, added in quadrature, to estimate the uncertainty in these fluxes
(see Fig. 7 and Sect. 5.2 for fractional uncertainties).  We note that the uncertainty in STILT
transport (see Sect. 4.1 for details) is inherently reflected in the relative uncertainty of the
regression coefficients ($a_1$, $a_2$…).
$Z = aHs + b + \xi$           (2)



$Z = a_1 Hs_1 + a_2 Hs_2 + (a_3 Hs_1 Hs_2) \ldots + b + \xi$                    (3)

We found statistically significant negative correlations between the upstream sea ice influence
and both $CHBr_3$ and $CH_2Br_2$ mixing ratios, and no positive relationships between upstream sea-
ice influence and any measured HVOC, such as $CH_3I$ in Region 1 (Fig. 7). Note, sea ice did not
include land ice; however, we also found a negative correlation between upstream land ice
influence and mixing ratios of HVOCs. We interpret this result to mean that increased
summertime sea ice acts either to reduce the production of HVOCs by blocking sunlight or as a
physical barrier to oceanic emissions of HVOCs from under-ice algae. Both of these mechanisms
are also consistent with a link between enhanced $CHBr_3$ and $CH_2Br_2$ emissions due to sea-ice
retreat. High concentrations of $CHBr_3$ have been linked to sea ice retreat and surface sea-ice melt
water (Carpenter et al., 2007). We note that over-turned first year sea-ice, which can expose
under-ice algae colonies to the air, likely still present a local source of $CHBr_3$, $CH_2Br_2$, or other
HVOCs to the MBL.
In other studies, it has also been proposed that sea ice could be an important source for $CHBr_3$
and other HVOCs, since high mixing ratios of $CHBr_3$ have been observed at the sea-ice and ice-
snow interface in the austral winter (Abrahamsson et al. 2018) and in under-ice algae in the
austral spring (Sturges et al. 1993). At present, CAM-Chem v1.2 with very short-lived halogen
chemistry does not include a regional flux of HVOCs over sea-ice covered waters in summer,
and our results do not indicate a need to include one. Our data, which were collected in January
and February, however, cannot assess the importance of sea ice as a source of HVOCs in other
seasons, such as winter or spring (Abrahamsson et al. 2018; Sturges et al. 1993). More field
campaigns are needed to further study the seasonality and regional strength sea ice related
HVOC emissions.
We observed a statistically significant positive correlation between the footprints of 8-day
satellite composites of the chl $a$ concentration, which is widely used as a proxy for near-surface
phytoplankton biomass, and mixing ratios of $CHBr_3$ and $CH_2Br_2$ in Region 1 (Fig. 8a and Fig.
8b). This finding corroborates previous findings from ship-borne field campaigns and laboratory
studies that have suggested a biogenic source for these two bromocarbons (e.g., Moore et al.,
1996; Hughes et al., 2013), and further substantiates the current CAM-Chem parameterization of
regional bromocarbon emissions using satellite retrievals of chl $a$ in polar regions. $CH_3Br$
mixing ratios were not significantly correlated with chl $a$ footprints (Fig. 8c). Although
potentially suggesting that marine phytoplankton and microalgae were not a strong regional
source of $CH_3Br$ during ORCAS, it is also possible that the relatively long lifetime of $CH_3Br$
precludes a definitive analysis of its origin based on chl $a$ using 7-day back-trajectories. Neither
$CHCl Br_2$ nor $CHBrCl_2$ were significantly correlated with chl $a$ composite footprints (data not
shown); however, more observations of these short-lived species in the remote MBL are needed
to substantiate this result.
Similar to Lai et al. (2011), we observed a significant correlation between mixing ratios of $CH_3I$
and total weekly upstream influence functions of 8-day chl $a$ composites (Fig. 8d). Weaker





correlations were observed with upstream influence functions on shorter timescales than seven
days. We found that $CH_3I$, particularly in Region 1, was better explained by a multi-linear
regression with two predictors: 1) the influence function of downward shortwave radiation at the
surface (Fig. 9a) and 2) the absorption of light due to detrital material (Fig. 9b), yielding
improved agreement between predicted and observed $CH_3I$ (Fig. 9c).
Although certain species of phytoplankton are capable of producing $CH_3I$ (e.g. Manley and de la
Cuesta 1997; Hughes et al., 2011), several studies also indicate a non-biological source for $CH_3I$
in the surface ocean. This non-biological source, though not fully understood, requires light, a
humic like substance at the surface ocean, and iron availability, which is scarce in the Southern
Ocean (Moore and Zarifou 1994; Richter and Wallace 2004). Iron, which is used extensively by
phytoplankton in the surface ocean, can be replenished in surface waters by wintertime mixing of
subsurface iron enriched waters, sea ice melt, intense recycling of organic material, and aeolian
dust (McGillicuddy et al. 2015; Tagliabue et al. 2014; Williams et al. 2007). Sources of iron that
may boost $CH_3I$ emissions other than recycling of organic material at the sea surface are an
omitted variable in our analysis.
Several previous studies have correlated mixing ratios of $CH_3I$ to satellite retrievals of
photosynthetically active radiation (PAR) and temperature, citing the link between temperature
and PAR to the solar radiation necessary for the photo-production of $CH_3I$ in surface waters (e.g.
Happell et al., 1996; Yokouchi et al., 2001). We note that chl *a*, which is a proxy for living algal
biomass, was correlated with CDOM in Region 1 and Region 2, ($r^2 = 0.24$; data not shown).
Finally, we note that photochemical loss during transport is not accounted for in this analysis.
Low OH mixing ratios, cold temperatures, and lower photolysis rates due to angled sunlight at
high latitudes lead to longer than average HVOC lifetimes. For instance, assuming an average
diurnal OH concentration of 0.03 pptv, and average photochemical loss according to the TUV
model and the Mainz Spectral data site (http://satellite.mpic.de/spectral_atlas) for Jan. 29 under
clear sky conditions at 60° S, $CHBr_3$ has a lifetime of 30 days, $CH_2Br_2$ has a lifetime of 270 days,
$CH_3I$ has a lifetime of 7 days, and $CHClBr_2$ has a lifetime of 63 days. As such, the
photochemical lifetimes of these gases are greater than or equal to the time of our back-trajectory
analysis. Moreover, OH concentrations in this region have large uncertainties, the inclusion of
which would lead to more, not less, uncertainty in geophysical influence function regression
coefficients and estimated fluxes.

**5 Flux estimation**
**5.1 $O_2$-based emission estimates**
We make use of the robust relationships between airborne observations of $O_2$ and HVOCs
combined with modeled $O_2$ fluxes to estimate HVOC fluxes over the Southern Ocean. For
$CHBr_3$, $CH_2Br_2$, and $CHClBr_2$ we construct ocean emission inventories for January and February
using a scaled version of modeled air-sea $O_2$ fluxes from simulations using a configuration of the
CESM model nudged to reanalysis temperatures and winds as described in Stephens et al. (2018)




to facilitate comparisons across regions and atmospheric models (Fig. 9).  An earlier free running
version of CESM was one of the best evaluated for reproducing the seasonal cycle of $O_2/N_2$ over
the Southern Ocean (Nevinson et al., 2015; 2016). To date, the north-south gradient in
atmospheric $O_2$ has not been well reproduced by any models (Resplandy et al., 2016). Vertical
gradients in $O_2$ on ORCAS indicate that CESM overestimated gradients by 47% on average;
accordingly, $O_2$ fluxes were adjusted downward by 47% to better match the observations.  This is
obviously a very simple adjustment to the modeled fluxes, and the actual air-sea $O_2$ flux biases in
CESM likely have a great deal of spatial and temporal heterogeneity. We calculated an
uncertainty for the CESM flux using a second, independent estimate of $O_2$ fluxes based on
dissolved $O_2$ measurements in surface seawater.  The Garcia and Keeling (2001) climatology has
much smoother spatial patterns than CESM flux estimates but also results in overestimated
atmospheric $O_2$ spatial gradients. We calculate the relative uncertainty in $O_2$ flux as the ratio of
the mean absolute difference between gridded Garcia and Keeling (2001; also adjusted down by
51 % to better match observations) to the CESM model flux estimates in Region 1 and Region 2
(adjusted down by 47%).  Based on the ratios of HVOC to $O_2$ mixing ratios in bivariate least
squares regressions and these adjusted $O_2$ fluxes, we estimate mean emissions of $CHBr_3$ and
$CH_2Br_2$ in Region 1 and Region 2.  Relative uncertainty in the slopes (i.e., the standard deviation
of the slopes) from these regressions and the mean relative uncertainties in regional $O_2$ fluxes
(7.3% in Region 1 and 3.4 % in Region 2) were added in quadrature to yield uncertainties in
calculated HVOC emission rates.

Figure 10 shows the mean emissions for Jan. and Feb. of $CHBr_3$, $CH_2Br_2$, and $CHClBr_2$ in
Region 1 and Region 2. Mean regional emissions of $CHBr_3$ and $CH_2Br_2$ and $CHClBr_2$ are 91 ± 8,
31 ± 17, and 11 ± 4 pmol $m^{-2}$ $hr^{-1}$ in Region 1 and 329 ± 23, 69 ± 5, and 24 ± 5 pmol $m^{-2}$ $hr^{-1}$ in
Region 2 (Table 1).  The mean flux of $CH_3I$ in Region 2 is 392 ± 32 (Table 1).  Table 1 also lists
the mean Jan. and Feb. CAM-Chem emissions from Region 1 and Region 2, as well as emissions
from several other observational and modeling Antarctic polar studies.  Our estimates fall within
the range of these other Antarctic polar studies, which span every month of the year and whose
estimated fluxes range from negative (i.e. from the atmosphere into the ocean) to 3500 pmol $m^{-2}$
$hr^{-1}$ $CHBr_3$ in a coastal bay during its peak in primary production.  CAM-Chem emissions for all
species are significantly lower than our observationally derived values in Region 1, with the
exception of $CH_3I$.  Conversely, CAM-Chem emissions are significantly higher than our
estimated biological emissions in Region 2, with the exception of $CHClBr_2$ in Region 1, which
remains under predicted by the model (Table 1).  We note that in Region 2, CAM-Chem fluxes
of $CHBr_3$ and $CH_2Br_2$, although still significantly different, are more similar to our estimated
fluxes.

**5.2 STILT-based emission estimates**
Similar to our $O_2$-based emission estimates, we used the relationship between geostatistical
influence functions and $CH_3I$ mixing ratios to predict a flux field in Region 1.  The shortwave



radiation and detrital material influence function coefficients and an interaction term from a
multi-linear regression (Fig. 9) were used to estimate an average non-biological flux of $CH_3I$
(Fig. 11; Table 1).  This method could be used in place of the current Bell et al. (2002)
climatology to update near weekly (~8 day) emissions of $CH_3I$ in future versions of CAM-Chem.
Our estimated regional mean flux in Region 1 (35 ± 29 pmol m$^{-2}$ hr$^{-1}$) is significantly lower than
the current CAM-Chem estimated emissions (Table 1).  As noted in Sect. 3, our observations of
$CH_3I$ are also much lower than the modeled mixing ratios. As discussed above, the strong
correlations between $CH_3I$ and $O_2$ in Region 2 also suggest a dominant biological source for this
compound.  As a result, we have not used this relationship to parameterize a flux for $CH_3I$ in
Region 2 (see Sect. 2.5 and 5.1 for details).

**6 Conclusions**
Our work combined TOGA and AWAS HVOC airborne observations from the ORCAS and
ATom-2 campaigns, with coincident measurements of $O_2$ and $CO_2$, geophysical datasets and
numerical models, including the global climate model CAM-Chem, and the Lagrangian transport
model, STILT.  We evaluated model predictions, calculated biogenic enrichment ratios, inferred
regional sources, and provided novel means of parameterizing ocean fluxes.  We found that the
Southern Ocean MBL is enriched in HVOCs, and these MBL enhancements are less pronounced
in Region 1 (at higher latitudes) than in Region 2 over the productive Patagonian shelf.  Our
results indicated that the Southern Ocean poleward of 60° S (Region 1) and Patagonian Shelf
(Region 2) are moderate regional sources of $CHBr_3$, $CH_2Br_2$, and $CH_3I$, and weak sources of
$CHClBr_2$ and $CHBrCl_2$ in January and February.  CAM-Chem provided a good foundation for
parameterizing HVOC emissions, particularly for $CHBr_3$ and $CH_2Br_2$ in Region 1 and Region 2.
Conversely, $CHClBr_2$ and $CHBrCl_2$ emissions were underestimated by a factor of two or three in
the model, while $CH_3I$ emissions were overestimated by a factor of more than three, and airborne
observations indicated that the CAM-Chem $CH_3Br$ surface boundary condition may be too low
by ~25%.
Our results suggested that summertime biological HVOC fluxes may be parameterized with
some success based on airborne observations of enrichment ratios, as well the influence of
remotely sensed parameters.  $CHBr_3$ and $CH_2Br_2$ exhibited strong and robust correlations with $O_2$
as well as weaker correlations with the influence of chl $a$, which is a proxy for phytoplankton
biomass.  $CHClBr_2$ and $CHBr_3$ were well correlated with one another.  Together, these
correlations suggested a biological source for these gases over the Southern Ocean.  We found
that $CH_3I$ mixing ratios in Region 1 were best correlated with a non-biological geophysical
influence function, although biogenic $CH_3I$ emissions appear important in Region 2.
Our flux estimates based on the relationship of HVOC mixing ratios to other airborne
observations and remotely sensed parameters compared relatively well with those derived from
global models and ship-based studies (Table 1).  Our emission estimates of $CHBr_3$, $CH_2Br_2$,
$CH_3I$, and $CHClBr_2$ were lower than most prior estimates from the Antarctic polar region in
summer, although they were significantly higher than CAM-Chem's prescribed emissions in
Region 1, where HVOC mixing ratios are under predicted (Table 1; Fig. 5). In the case of $CH_3I$,
our estimated emissions suggest that the prescribed emissions in CAM-Chem may be too high.





Our parameterization of the flux in Region 1 and Region 2 could be used to explore inter-annual
variability in emissions, which is not captured by the Bell et al. (2002) climatology currently
employed in CAM-Chem.
To extend these relationships to year-round and global parameterizations for use in global
climate models, they must be studied using airborne observations in other seasons and regions.
Nevertheless, these methods may facilitate parameterizing emissions of new species or
improving existing emissions. Finally, future airborne observations of HVOCs have the
potential to further improve our understanding of air-sea flux rates and their drivers for these
chemically and climatically important gases over the Southern Ocean.
*Data Availability*. The ORCAS and ATom-2 datasets are publically available at
(https://doi.org/10.5065/D6SB445X ; www.eol.ucar.edu/field_projects/orcas) and
(https://doi.org/10.3334/ORNLDAAC/1581).
*Author Contributions*. EA is responsible for the bulk of the conceptualization, formal analysis,
writing, review, and editing with contributions from all authors. BBS and ECA were
instrumental in the investigation and supervision related to this manuscript. RSH contributed to
the conceptualization, as well as the investigation and HVOC data curation for this project. BBS,
EJM, and RFK were responsible for the data curation of $O_2/N_2$ data and contributed to formal
analysis involving these data. MSHM along with EAK were responsible for STILT data curation
and formal analysis, and the conceptualization and formal analysis of SITLT-based geostatistical
influence functions and flux estimates were also informed by these two. DK, along with ST, JFL
and ASL were responsible for constructing CAM HVOC emissions and conducting CAM runs.
MCL was responsible CESM simulations yielding $O_2$ fluxes and comparing this product
alongside the Garcia and Keeling $O_2$ climatology in CAM. KMC and CM were responsible for
the data curation of $CO_2$ observations. AJH contributed to the investigation for HVOC data.

*Acknowledgements*. We would like to thank the ORCAS and ATom-2 science teams and the
NCAR Research Aviation Facility and NASA DC-8 pilots, technicians and mechanics for their
support during the field campaigns. In addition, we appreciate the NCAR EOL staff who have
facilitated computing and data archival. In particular, we thank Tim Newberger for his help in
supporting the NOAA Picarro $CO_2$ observations and Andrew Watt for his help in supporting the
AO2 $O_2$ observations. This work was made possible by grants from NSF Polar Programs
(1501993, 1501997, 1501292, 1502301, 1543457), NSF Atmospheric Chemistry Grants
1535364, 1623745, and 1623748 and NASA funding of the EVS2 Atmospheric Tomography
(ATom) project, as well as the support of the NCAR Advanced Study Program (ASP)
Postdoctoral Fellowship Program and computing support from Yellowstone, provided by
NCAR's Computational and Information Systems Laboratory. The National Center for
Atmospheric Research is sponsored by the National Science Foundation.





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





**Tables**
Table 1. HVOC emission estimates (pmol $m^2$ $hr^{-1}$) in Region 1 and Region 2 calculated in this
study, from CAM-Chem (Ordoñez et al. 2012) and from several other modeling and ship-based
observational studies.

| Region/Months | CHBr$_3$ | CH$_2$Br$_2$ | CH$_3$I | CHClBr$_2$ | Reference |
|---|---|---|---|---|---|
| Region 1 (JF) | 91 ± 8 | 31 ± 18 | 35 ± 29 | 11 ± 4 | This Study |
| Region 2 (JF) | 329 ± 23 | 69 ± 5 | 392 ±32 | 25 ± 5 | This Study |
| Region 1 (JF) | 10 | 1.9 | 120 | 0.38 | CAM-Chem |
| Region 2 (JF) | 360 | 44 | 800 | 8.7 | CAM-Chem |
| Southern Ocean (≥50°S), (DJ) | 200 | 200 | 200 | -------------- | Ziska et al. 2013 (model) |
| Marguerite Bay (DJF) | 3500 | 875 | -------------- | -------------- | Hughes et al. 2009 (obs) |
| 70°S-72°S Antarctica | 1300 | -------------- | -------------- | -------------- | Carpenter et al. 2007 (obs) |
| Southern Ocean (≥50°S) (Feb. - April) | 225 | 312 | 708 | -------------- | Butler et al. 2007 (obs) |
| 40°S-52°S S. Atlantic (Sept.-Feb.) | -1670 | -------------- | 250 | -------------- | Chuck et al. 2005 |
| Southern Ocean (≥50°S), (DJ) | -330 | -------------- | -------------- | -------------- | Mattson et al. 2013 (model) |






**Figures**

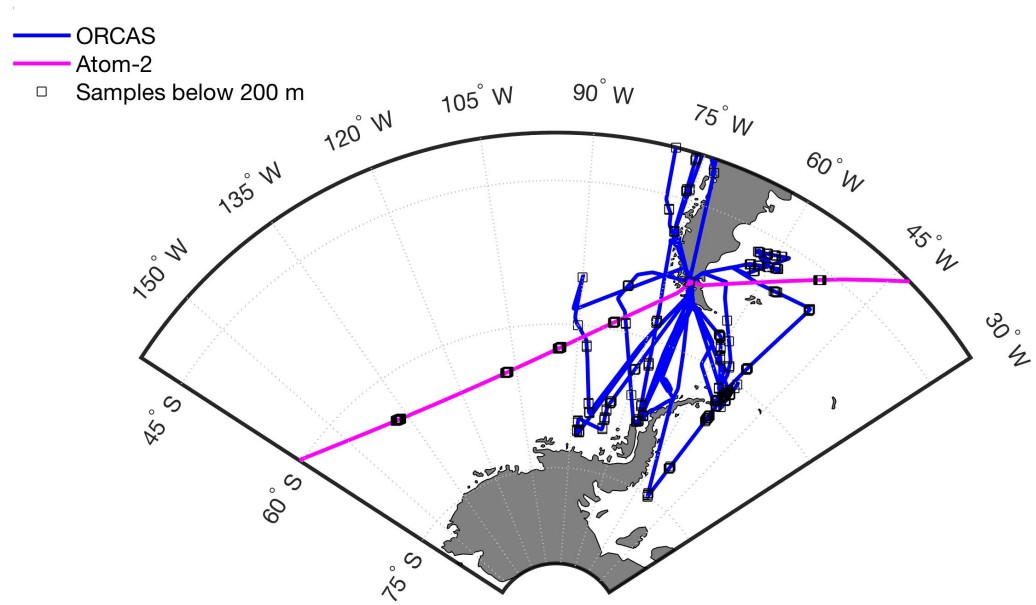

**Figure 1.** Overview map ORCAS and ATom-2 flight tracks in the study regions: 1) high latitudes in the Southern Hemisphere poleward 60° S and 2) the Patagonian Shelf. The ORCAS and ATom-2 aircraft flights and dips below 200 m that took place within these regions are also shown.



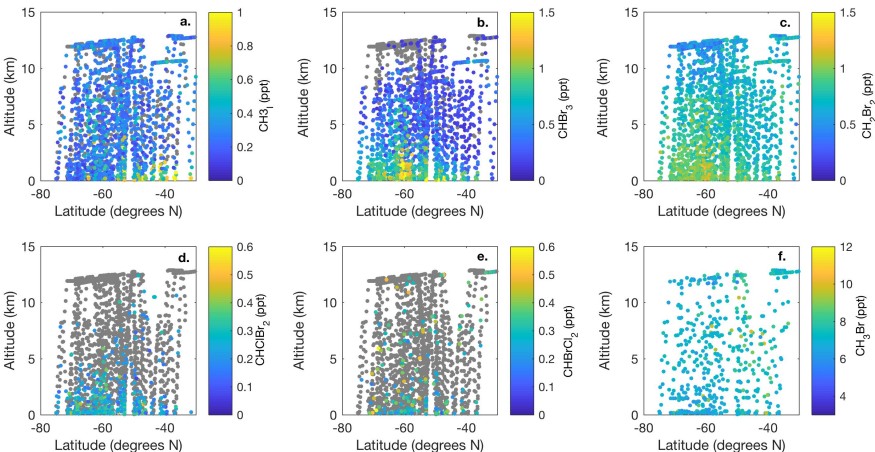

924

**Figure 2.** Meridional-altitudinal cross-sections of mixing ratios of a) $CH_3I$ , b) $CHBr_3$, c)
$CH_2Br_2$, d) $CHClBr_2$, and e) $CHBrCl_2$ from the TOGA and mixing ratios of f) $CH_3Br$ from
AWAS and WAS in 2016 and 2017, respectively, during the ORCAS and ATom-2 campaigns
over the Southern Ocean in the austral summer. Note the different color bar scales. Gray points
denote measurements below the detection limit of each species, respectively.







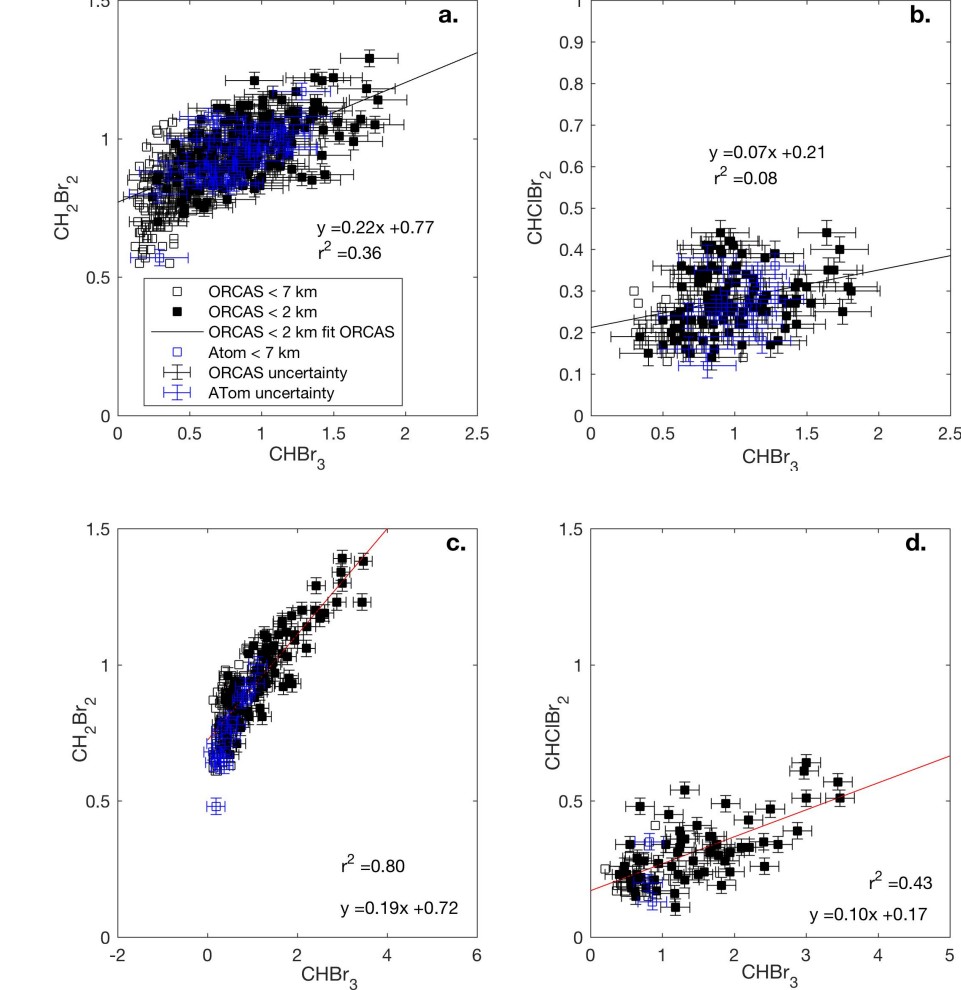





**Figure 3.** Mixing ratios of $CHBr_3$ vs. $CH_2Br_2$ across the ORCAS and ATom-2 campaigns in Region 1 (Fig.3a,b) and in Region 2 (Fig.3c,d). Type II major axis regression model (bivariate least squares regressions) are based on ORCAS data below 2 km illustrates a regional enhancement ratio. Error bars represent the uncertainty in HVOC measurements.

940

941





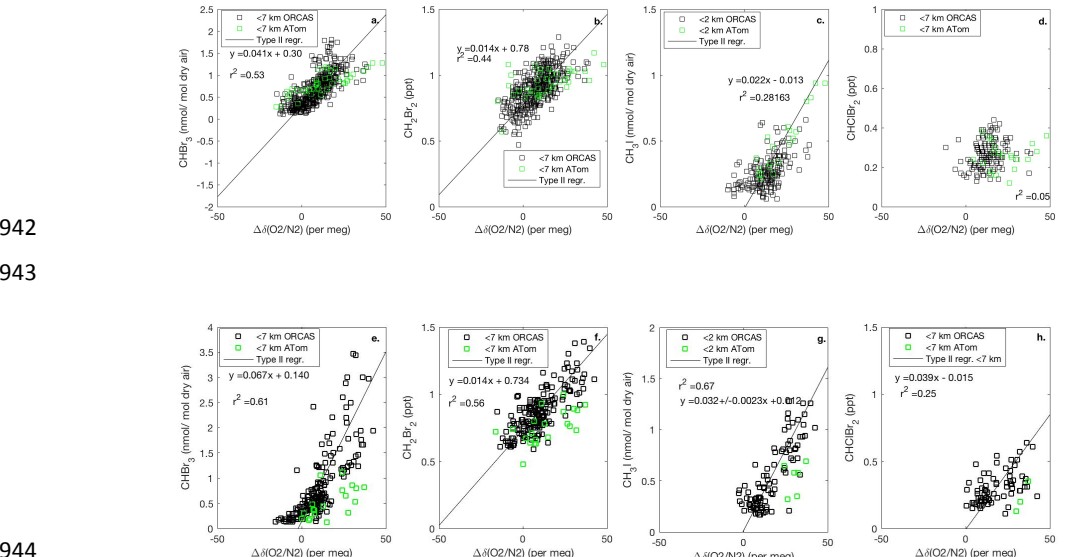

**Figure 4.** Mixing ratios of $CHBr_3$, $CH_2Br_2$, and $CH_3I$ vs. $O_2$ on ORCAS and ATom-2 in Region 1, pole ward of 60° S (a-c) and Region 2 over the Patagonian Shelf (e-f). Slopes ± standard errors from type II major axis regression model (bivariate least squares regression) fits of ORCAS data (using variables scaled to their range) are shown. To isolate the contribution of ocean $O_2$ fluxes, the ORCAS $\delta(O_2/N_2)$ values reported here represent the $\Delta\delta(O_2/N_2)$ to observed values between 5-7 km and are adjusted for CESM $O_2$ land and fossil fuel contributions and the influence of air-sea $N_2$ fluxes. Figure S3 shows the same plots relative to unadjusted ORCAS $\delta(O_2/N_2)$ values and illustrates that these corrections are minor. The slopes reported in the figure are converted to pmol:mol ratios prior to estimating biogenic HVOC fluxes based on modeled CESM $O_2$ fluxes. Data from above 7 km were excluded due to the influence of air masses transported from further north.



958

959

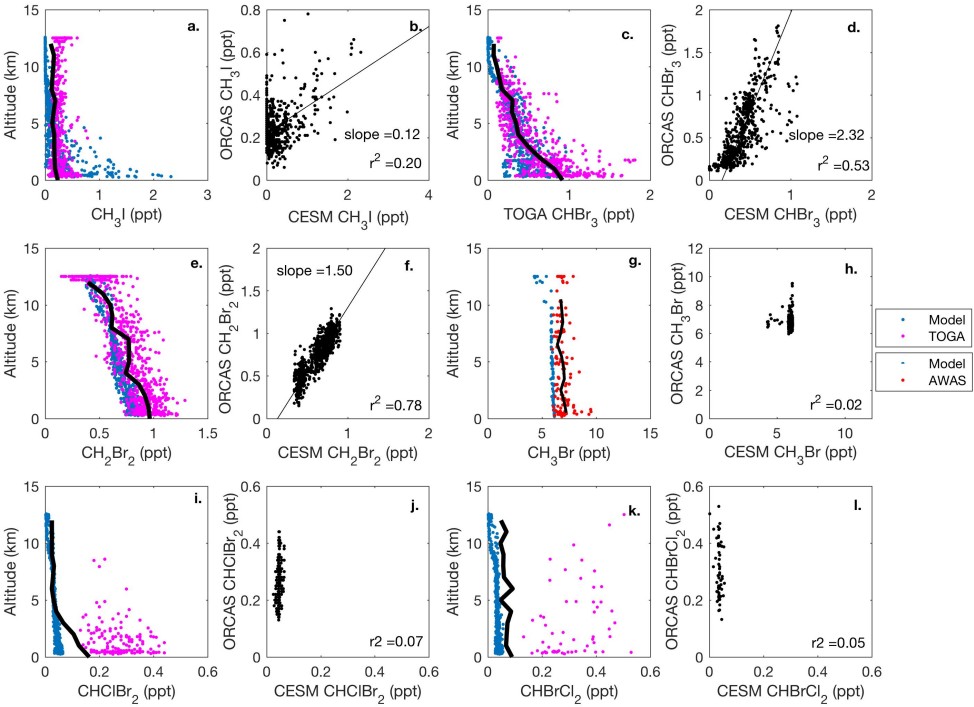

960

**Fig 5.** CAM-Chem1.2 model-aircraft measurement comparison during the ORCAS campaign
between 1-12 km in Region 1, high latitudes in the Southern Hemisphere poleward 60° S. All
regressions are type II major axis regression models bivariate least squares regressions (slopes
are shown when the $r^2 \geq 0.2$). The bold, black line in each vertical profile represents the binned
(mean) mixing ratio of HVOC measurements at that altitude, including measurements below the
detection limit (DL), which are assigned a value equal to the DL multiplied by the percentage of
data below detection. Modeled values include locations where observations were below the DL.





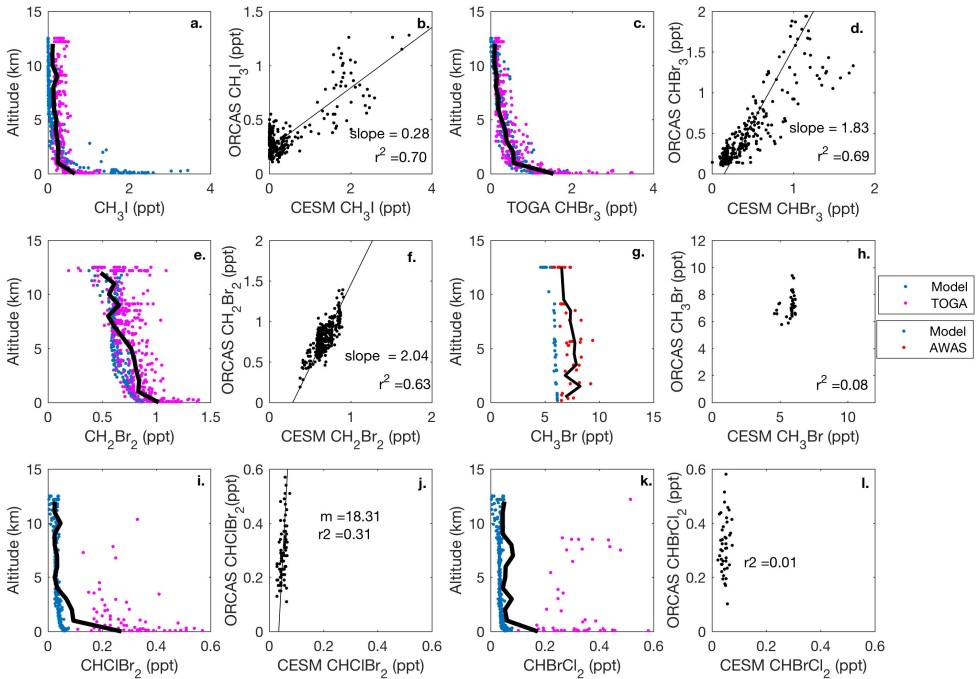

968

**Figure 6.** CAM-Chem 1.2 model-aircraft measurement (TOGA and AWAS) comparison during
ORCAS campaign between 1-12 km in Region 2, the Patagonian Shelf. All regressions are type
II major axis regression models bivariate least squares regressions (slopes are shown when the $r^2$
≥ 0.2). The bold, black line in each vertical profile represents the binned (mean) mixing ratio of
HVOC measurements at that altitude, including measurements below the detection limit (DL),
which are assigned a value equal to the DL multiplied by the percentage of data below detection.
Modeled values include locations where observations were below the DL.

976

977





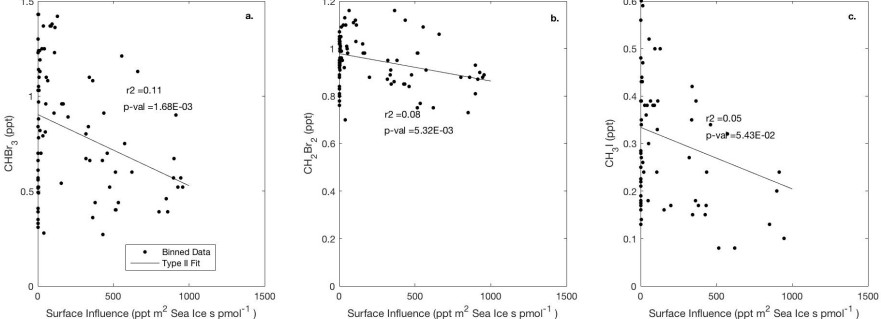

978

**Figure 7.** Linear type II regressions between influence functions convolved with sea ice
distributions (not includind land ice), and mixing ratios for $CHBr_3$, $CH_2Br_2$, and $CH_3I$ in Region
1, poleward of 60° S. Surface influence functions (ppt $m^2$ s pmol$^{-1}$) in each grid cell were
multiplied by predictor variables, such as fractional sea ice concentration, which is unit-less,
yielding sea ice surface influence units of ppt $m^2$ s pmol$^{-1}$, as shown on the x-axis. Linear
regression lines are only shown where a statistically significant relationship was found.

985





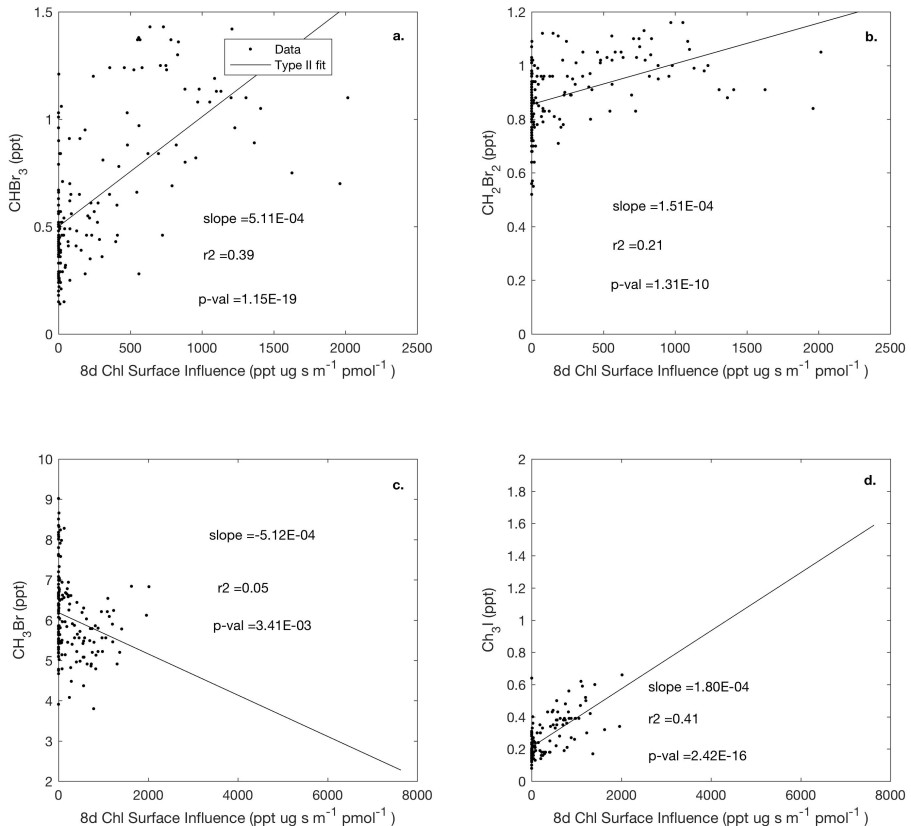

986

987

**Figure 8.** Linear type II regressions between influence functions of eight day composites of chl
*a* and mixing ratios of HVOCs (a-d) poleward of 60° S (Region 1). Surface influence functions
(ppt $m^2$ s $pmol^{-1}$) in each grid cell were multiplied by predictor variables, such as chl. *a* ($\mu g\ m^{-3}$),
resulting units of ppt s $pmol^{-1}$ $m^{-1}$, shown on the x-axis. Linear regression lines are only shown
where a statistically significant relationship was found.

993





994

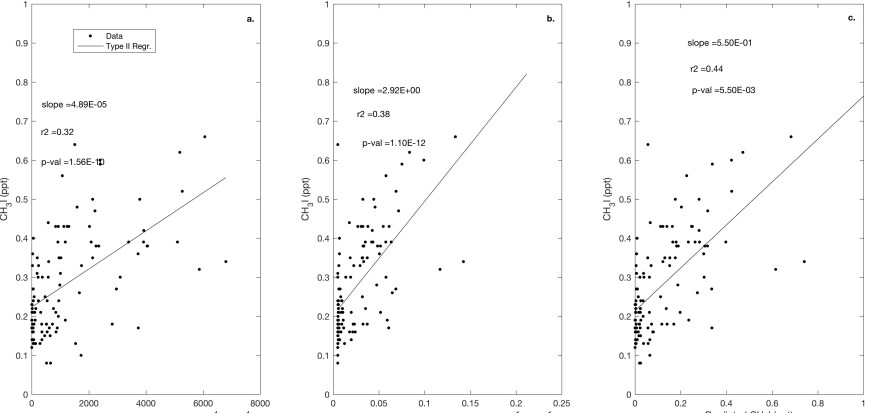

995

**Figure 9.** Observed CH$_3$I plotted against geostatistical influence of downward shortwave
radiation (a) and absorption due to detritus (b) and the predicted mixing ratios of CH$_3$I based on a
multiple linear regressions (MLR) using these two predictors and an interaction term in Region
1, poleward of 60° S (c).  Surface influence functions (ppt m$^2$ s pmol$^{-1}$) in each grid cell were
multiplied by predictor variables, such as shortwave radiation (W m$^{-2}$), yielding units of ppt Ws
pmol$^{-1}$, and detrital absorption (m$^{-1}$), yielding units of ppt m s pmol$^{-1}$, shown on the x-axes.
Based on these relationships (a,b), we included these predictors in a multiple linear regression (±
standard deviations; Equation 2), with an intercept b = 0.19 ± 0.01, and influence coefficients a$_1$
= 3.7E-5 ± 1.3E-5, a$_2$ = 3.5± 0.74, and an interaction term with the coefficient -5.2E-4 ± 1.5E-4
(c).

1006





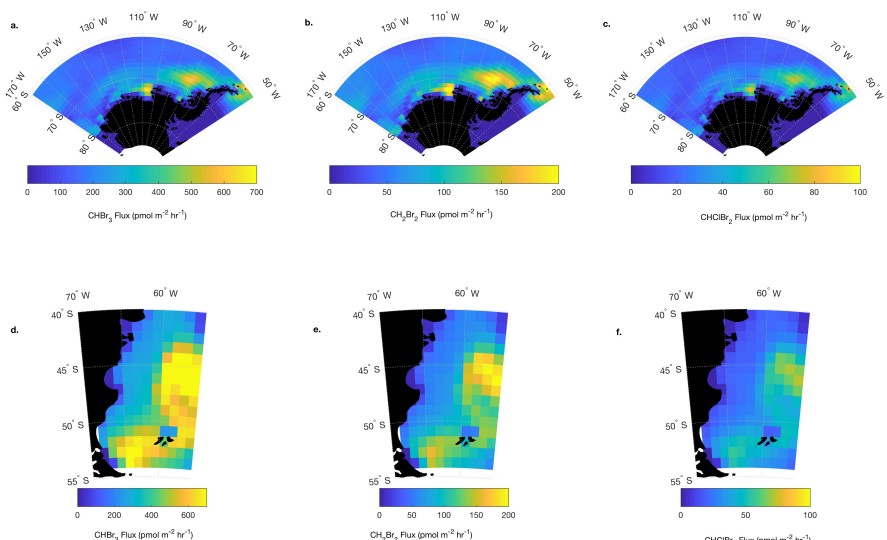

1007

**Figure 10.** Mean Jan. – Feb. $O_2$-based $CHBr_3$ and $CH_2Br_2$ and $CHClBr_2$ fluxes (pmol m$^{-2}$ s$^{-1}$) in Region 1 (a-c) poleward of 60° S and Region 2 (d-f) over the Patagonian Shelf. CESM $O_2$ fluxes are scaled by the slope between the oceanic contribution to $\delta(O_2/N_2)$ and $CHBr_3$ and $CH_2Br_2$, and and $CHClBr_2$ reported in Fig. 4. Note that these fluxes represent mean biogenic fluxes in Jan. - Feb. (see Sect. 5.1 for details).

1013





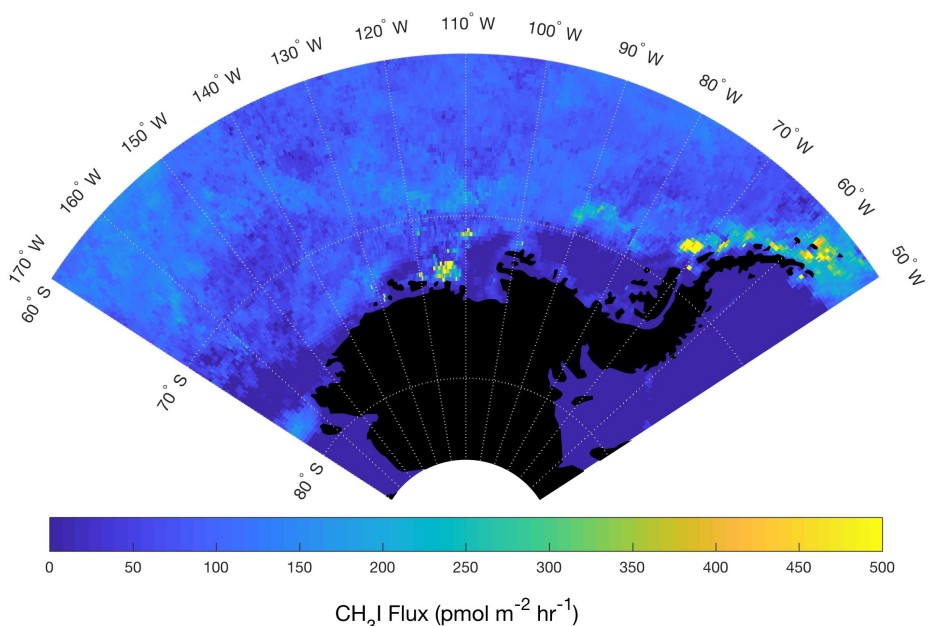

1014

**Figure 11.** Mean estimated $CH_3I$ fluxed for Jan. – Feb. The multilinear regression in Fig. 9
between $CH_3I$ mixing ratios and geophysical influence functions related to shortwave radiation
and detrital material at the sea surface was used to derive a mean flux field in Jan.-Feb., 2016 for
Region 1.

1019

1020