# Peer review of "Novel approaches to improve estimates of short-lived halocarbon emissions during summer"

_Atmospheric Chemistry and Physics, 2019_

## Referee Comment (RC1) · Anonymous Referee #1 · 25 Apr 2019

This study of airborne observations of halogenated VOCs (HVOCs) represents a valuable addition to the knowledge of these compounds over the Southern Ocean, where few data exist. The study confirms the current view that the main sources of $CHBr_3$ and $CH_2Br_2$ are biological, and that $CH_3I$ has both biological and non-biological sources. The authors have put forward a novel concept of using enrichment ratios of HVOCs to $O_2$ to infer the contribution or otherwise of ocean biological sources, and propose a new function to estimate non-biological emission fluxes of $CH_3I$. The dataset has been used to evaluate the CAM-Chem HVOC emission scheme at high latitudes in the Southern Hemisphere. The take home message/s from this evaluation are rather opaque – they could do with being put in context. E.g., do they infer that fluxes from these regions

are poorly known, or problems with the models mixing /convection schemes special to these latitudes, or issues with photo-oxidation rates?.

In terms of presentation, the paper has a number of typographical and other errors, listed below, and needs a thorough reading (I doubt I captured all of them).

However overall, I think this manuscript presents sufficiently novel results to be suitable for publication, once these matters have been attended to.

Specific comments:

Ln 26 onwards. The regional enrichment ratios should be put in context here - there is no explanation as their relevance.

Ln 50-52 "Indeed, HVOCs may be among the most important sources of inorganic bromine to the whole atmosphere ..... (Murphy et al.,in review)."

This is not conventional wisdom and thus quite a bold statement. Are the authors confident that the Murphy et al paper will be published soon?

Lns 60-61. The anthropogenic sources of CH3Br have changed over time and now are dominated by quarantine and pre-shipment (QPS) applications (not controlled by the Montreal Protocol). Please stick to the most recent information from WMO 2018 (and update the reference).

Lns 110-117. The last paragraph of the introduction would benefit from an introduction to the concept of enrichment ratios of HVOCs to O2, which feature prominently in the abstract.

Lns 204-218. The fact that the polyhalogenated bromocarbons are likely co-emitted is not new – there are numerous papers that show this, and the discussion could elaborate on those a bit more. What is also missing from this paragraph is a discussion of macroalgal sources of these compounds, although this is presumably not relevant for the Antarctic.

Lns 213-214 "For instance, Huges et al. (2013) also report distinct seawater slopes between CH2Br2 to CHBr3 , when chl a was increasing." It is not clear what is meant by this. Please rephrase.

Lns 312- 313. "In both regions, the model under predicts CH3I above the MBL, which may indicate slower observed photochemical loss than the model predicts." Has this been found in other CAM-Chem studies – e.g. is it a general result? If not, could a different source emission distribution (i.e. more homogeneous source) explain these results?

Ln 468 onwards. There is no mention in Moore and Zarifou 1994 nor Richter and Wallace 2004 as far as I can see on the influence of iron availability – do the authors mean iodide availability?!

Lns 901 onwards (Table 1). Note that units should be pmol m-2 hr-1 (not m2). Please state whether the values given for the observations are means or medians. It would be also be good to include their ranges.

Typos:

Ln 82. atmopsheric Ln 213. "oppose" should be "opposed" Ln 213. "Huges" should be "Hughes" Ln 242 : "HOVCs" Ln 469. "Zafarou" should be "Zafariou" Ln 980. "includind" LN 1015. "fluxed"

---

## Referee Comment (RC2) · Anonymous Referee #2 · 1 May 2019

The paper presented by Asher et al., provides a valuable contribution to the understanding of the distribution and sources of halogenated organic substances from the Southern Ocean. The work publishes airborne observations of VHOC in an understudies region and applies new concepts for source determinations related to measurements of O2 and CO2 and geophysical datasets. It underlines current knowledge of the biological sources of CHBr3 and CH2Br2 by applying their ratio to oceanic oxygen emissions. CH3I appears to have a dominant biological source in the area of the Patagonian shelf, while closer to Antarctica a photochemical source appears to be dominant. The paper also compares the derived emissions of the novel concepts to the output of a global climate model.

I agree that the presentation of data from several compounds and several campaigns is a difficult task. Also the loaded content of the paper: evaluation of model predictions, calculation of biogenic enrichment ratios, identification of regional sources, and novel means of parameterizing ocean fluxes, which was only summarized clearly in the conclusion section of the paper, makes the task of writing not easier. While the authors present their results and outcomes, which are not totally exciting and sometimes are also not very convincing due to poor correlations, their novel approaches and novel concepts are more exciting, but are poorly presented. They could do a much better job in explaining and presenting their concepts and the overall goal of the paper, which for me remains more a concept than a result paper. The results underline the novel approaches, as they do not contradict earlier studies and the novel approaches can be more useful and should be tested for and in future studies.

The authors should think about a different setup of their paper, putting their concepts more into the focus, but clearly their approaches need be described more clearly and in more detail throughout the text. Also they authors should think about the title. Also technically the paper needs improvement, as abbreviations are sometimes not introduced, sometimes edits are not clearly overworked, which led to typos and grammar mistakes and also I wonder if it would be possible to make some sentences less bulky and loaded. Some figures are too small, some legends appear odd and there appear to be misunderstandings with some references .

Overall I think the work behind the paper is very valuable and should be published in ACP, but the presentation of the work needs prior strong improvement.

Detailed comments below:

23-24: We also use CH3Br from the University of Miami Advanced Whole Air Sampler(AWAS) on ORCAS and from the UC Irvine Whole Air Sampler (WAS) on ATom-2.

In connection with the first and the third sentence, this is a strange sentence. I think there is too much detail in the first two sentences about the instrumentation , which

could be abandoned for the abstract and only explained later in the text.

37: Based on these relationships

What does this refer to?... it is unclear

49-50: Indeed, HVOCs may be among the most important sources of inorganic bromine to the whole atmosphere, since recent evidence indicates that sea salt is scarce..

This is not true, as there is enough literature out to show how sea salt aerosol dominates the bromine in the lower troposphere. If the authors want to keep this sentence they have to provide more evidence, than just an upcoming paper. I suggest rewriting and specifying the statement to the known literature.

69 to 74: There is an important observational paper missing which the authors need to relate to in the discussion of their results later on in the paper. It is: Regional sinks of bromoform in the Southern Ocean from 2013 from Mattsson et al. in GRL, where he shows the heterogeneity of the sources, which make the ocean a sink at times. Therefore also the next sentence needs to be revised: These studies indicate moderate ocean sources of CHBr3 and CH2Br2 at high latitudes in the Southern Hemisphere, and refer to Mattsson., possibly in line 79.

86 to 94: Here you need to relate to Salawitch (2011?).. –most trace gases in tropospheric air enter the stratosphere in the tropics, move poleward and descend to the troposphere at middle and high latitudes. Salawitch claims, that the polar bromine can be influenced by large scale subsidence from the lower latitudes...

97-98: Few constraints on HVOC mixing ratios or emissions based on airborne data exist at high latitudes in the Southern Hemisphere. What does this mean?

108: This is pmol... not nmol mol -1

110 to 117: Here you could (you need to do it somewhere) elaborate on the O2/N2

concept and why you chose to relate the HVOHCs to those.

124 to 131: Please include the regions into Figure 1.

126-128: If you only refer to two flights from ATom-2, the sentence could be easier to read.

174: Model is missing

177: You did not introduce CAM.

185: what is a broadening effect?

211-212: We note that the non-linearity observed in ratios of these two gases at low CHBr3 levels likely reflects the differences in emissions during strong phytoplankton blooms, as oppose to other periods.

The ratio may simply (and more likely) reflect other air masses from more distant source regions, which is reflected in a ratio which favors the longer lived compound ( CH2Br2) over the shorter- lived compound (CHBr3) which is emitted in larger quantities in a biological source region (refer to Yokouchi, 20xx) but more rapidly degraded during transport

213-214: For instance, Huges et al. (2013) also report distinct seawater slopes between CH2Br2 to CHBr3 , when chl a was increasing.

This is a weak sentence; can you give it more meaning?

219 . . .: please explain the concept: What do you expect from the ratio of the HVOHCs and the marine oxygen.  238/241-242: Where did you get this equilibration times? Support them by reference or evidence. And also the air-sea fluxes of O2 and CHBr3 are not very similar. Revise.

244-253: This paragraph is a little back and forth between compounds and regions; it can be sorted for easier reading.

[Figure]

263...: This should have come earlier, when you start with the equilibria (238, 241). And do you also reference the atmospheric lifetimes?

284: FINN and MEGAN 2.1 products. I guess the abbreviations need to be explained a bit as well as the products

286: from this sentence it is not clear where the oceanic emissions are derived from. I guess its Ordonez, 2012?

289: Ordonez, 2012 does not include CH3Br. Revise

334: GDAS has to be introduced.

353-354: We consider the wind direction error to evaluate the possible size of spatial errors in footprint location.

There appears to be something wrong with the grammar? The sentence is not understandable.

355-356: Given median wind speeds in this domain, this corresponds to a possible error of 260 km/day possible error.

Here is also something wrong.

375 and 376: OCI and GIOP have to be introduced. What does . . ..and its uncertainty . . .mean?

382: how do you obtain a 0.25° x 0.25° gridded uncertainty in the detrital material absorption? It is also not clear from section 5.2.

394: is the new terminology geophysical influence function something different than the surface influence function? Or why do you change the wording? Its unclear.

403 to 404. Can you give an example for H and s. What is the potential geophysical source distribution s?

412: here the potential source distributions is Hs1, Hs2. . .? And not s? Is HS1 ( line

406 ) the same as Hs1?

415-416: We used the standard deviation of the regression coefficients and the relative uncertainty in the source fields, added in quadrature, to estimate the uncertainty in these fluxes (see Fig. 7 and Sect. 5.2 for fractional uncertainties).

418: How did you calculate and do you report the relative uncertainty of the regression coefficients?

There is no standard deviation of the regression coefficients in Fig 7 and sect 5.2 does not explain fractional uncertainties and no explanation is found about relative uncertainties in source fields. Or are you relating to surface influence strength uncertainty here. There needs to be more explanation about this added here.

424: why did you include... such as CH3I in Region 1? The second half sentence does not add information?

425-426: Note, sea ice did not include land ice; however, we also found a negative correlation between upstream land ice influence and mixing ratios of HVOCs.

Why do you add the sentence starting with however? How did you get the correlations when it is not included and does it help the interpretation of the results? It appears misleading and redundant.

432-433: We note that over-turned first year sea-ice, which can expose under-ice algae colonies to the air, likely still present a local source of CHBr3, CH2Br2, or other VOHCs to the MBL.

How does this speculation relate to your study and how does it help your interpretations? It stands a bit loose currently.

461-464: What was the temporal resolution of the input data shortwave and detrital material- add in section 4.2.

468-474: This section is wrong. There is no study ( at least not the referenced ones)

[Figure]

which proves a relationship between iron availability and methyl iodide. The authors have misinterpreted the cited studies. Please check and revise.

476: citing the link between temperature and PAR to the solar radiation..this wording is strange..also add which temperature is needed. . .water . may be its easier to just write revealing the link to solar radiation ..or similar

483: please introduce TUV

480 to 490: this section appears to beat the wrong place. I would expect this earlier in the description of the model, e.g. in 4.1., where you also talk about uncertainties due to meteorology.

494-495: as explained earlier the concept needs to be introduced more clearly earlier..e.g. why do you not take the VOHC directly but apply their relationship to oxygen?

502 -518: How can a model and an observation based flux-estimate be wrong by around 50%? And why do you think that a simple down scaling of the calculated oxygen fluxes leads to a robust flux estimate for VOHC, respectively why is this better, than taking just the VOHC fluxes? Can you explain this concept in the text please? Also it is unclear why you calculate all your influence functions to the VOHC mixing ratios directly, and not to their relation to oxygen and why for the flux calculation this now appears better?

540: I strongly believe that the calculation of the regression surface influence functions need to be shown in the text not in the legend of figure 9.

543: You need to indicate in table1 ,which method you used in "This study" to derive the reported flux, as there are several methods here.

564-565: this also appears true for CHBr3, CHClBr2 in region 1 ..and for the entire troposphere for CHBrCl2

578: although they were significantly higher than CAM-Chem's prescribed emissions

in Region 1, where VOHC mixing ratios are under predicted (Table 1; Fig. 5).

Can you please add the comparison to CAM- Chem at the beginning .It would be better structured if you don't jump between comparisons.

581: parameterizations..these are different ..and you need to add which compounds you are referring to in this sentence.

586-587: Nevertheless, these methods may facilitate parameterizing emissions of new species or improving existing emissions.

Here it would be good to extend on the methods and why they are appear so usefull.and how you would extend them to other species.

Figure 2: Are the data of the campaigns merged? Detection limits need to be added. The label of CH3I is odd.

Figure 3: Please specify one name for the campaigns and keep it. Here in one figure the authors switch between Atom-2, Atom and Atom. Line 937 to 939 in the legend : This sentence does not make sense.

Figure 4. There appears to be an old legend as d, g and h are missing as well as CHClBr2. The applied regressions appear to be the same , thus it would be good to elaborate in the text about the method to reduce the legend, e.g. what means. using variables scaled to their range? In the legend? Also here only regressions above 0.2 are shown .

Figure 5,6: Switching between CESM in the figure and CAM-Chem1.2 does not help clarity. . . .multiplied by the percentage of data below detection.. .. was it used for calculating the mean? ..rephrase for clarity.

Figure 7: Talking of statistical significance with r2 « 0.2 and looking at the plots with scattered values and no surface influence, is a bold exaggeration. And the p-values can be abandoned from the figure and just the threshold mentioned, as they do not

help the statistics.

Figure 9: The labeling of the figures is too small, the p- value redundant and the legend for figure c.) too intricate. I strongly believe that the calculation of the regression surface influence functions need to be shown in the text not in the legend of figure 9.

Figure 10: The figures and labeling of a to c are too small. ( I suggest single plots, resolution as Figure 1?. It must be pmol m-2 hr-1 also in the legend. Also clarify that these are model results. How do the mentioned CESM (CAM- Chem 1.2) O2 fluxes relate to the figure? And is this also 2016?

Figure 11: fluxes not fluxed

––––––––––––––––––––––––––––––

---

## Referee Comment (RC3) · Anonymous Referee #3 · 21 May 2019

The manuscript of Asher et al. describes airborne observations of halogenated volatile organic compounds over the Southern Ocean and improved emission flux estimates, based on modelling studies and correlative O2 observations. This is an important and interesting study that should be published in Atmos. Chem. Phys. after consideration of the following points. The authors should consider improving the presentation by first presenting their data and methods and then discussing the results. This study contains important new methods and approaches compared to previous studies but the presentation is not always clear. As an example, a key result is the presentation of "regional enrichment ratios" for HVOCs, but it did not become sufficiently clear to me, how they are defined and how they were calculated.

[Figure]

Specific comments:

L32-34: in the same sentence "halogenated hydrocarbon" and "halogenated volatile organic compounds (HVOCs)" are used. If the two mean the same, use only one name. If there is a distinction, please define.

L47-49: Is there a particular logic for the order of the citations given? They are neither sorted according to year, nor alphabetically.

L50: "recent evidence indicates that sea salt is scarce and insufficient": this is a strong statement that should be backed up with more than a manuscript in review.

L66: You may cite Abrahamsson et al. (2018) already at this stage.

L96: The point "support quantitative air-sea flux estimates" is less obvious than the other points so a reference may be helpful here.

L211: "We note that the non-linearity observed in ratios of these two gases at low CHBr3 levels likely reflects the differences in emissions during strong phytoplankton blooms, as oppose to other periods." Could not the different lifetimes also effect this?

Fig. 3. Units missing for the axes

Fig. 4. Why are some units given as nmol/mol and others as ppt ?

L222: Sorry, but I don't know what a type II major axis regression is. A few more words may help.

L250: Please explain how the molar enrichment ratios are defined and/or calculated. This seems to be critical, but not well explained. Is this just the slope of the regression between CHBr3 (or CH2Br2) and O2?

L351: "In its simplest approximation, the wind speed error will correlate with surface influence error" I understand that this is in general may be a reasonable assumption, but it is not obvious to me why the error in the influence function (in ppt m2 s pmol-1)

should be proportional to the error in wind speed. More justification of this argument would be needed here.

L389: PAR: please spell out (as far as I can see first defined in L476)

L431: "We note that over-turned first year sea-ice, which can expose under-ice algae colonies to the air, likely still present a local source of CHBr3, CH2Br2, or other HVOCs to the MBL." What is this statement based on?

L499: Reference to Fig.9 in L499 was not clear to me. Was really Fig.9 meant here?

Fig. 9c: Caption not very clear, would be helpful if the description in the caption can be improved.

5.2 Why are STILT based emission estimates presented only for CH3I? Why is it not possible to perform this for other HVOCs as well?

Figure S4: "Consecutive samples in and out of dips into the MBL": Sorry, I don't really understand what is meant here, please re-word.

Technical corrections:

L134: "low attitude" -> "low altitude"

L183: citation should be part of the sentence

---

## Author Response (AR4)

Responses to reviewer #1

This study of airborne observations of halogenated VOCs (HVOCs) represents a valuable addition to the knowledge of these compounds over the Southern Ocean, where few data exist. The study confirms the current view that the main sources of CHBr3 and CH2Br2 are biological, and that CH3I has both biological and non-biological sources. The authors have put forward a novel concept of using enrichment ratios of HVOCs to O2 to infer the contribution or otherwise of ocean biological sources, and propose a new function to estimate non-biological emission fluxes of CH3I. The dataset has been used to evaluate the CAM-Chem HVOC emission scheme at high latitudes in the Southern Hemisphere. The take home message/s from this evaluation are rather opaque – they could do with being put in context. E.g., do they infer that fluxes from these regions are poorly known, or problems with the models mixing /convection schemes special to these latitudes, or issues with photo-oxidation rates?. In terms of presentation, the paper has a number of typographical and other errors, listed below, and needs a thorough reading (I doubt I captured all of them). However overall, I think this manuscript presents sufficiently novel results to be suitable for publication, once these matters have been attended to.

We appreciate the reviewer's time and comments. We have done our best to clarify the goals and findings of this study. We argue that emissions of HVOCs over the Southern Ocean are poorly known using mixing ratio comparisons with a global climate model and state of the art biogenic flux parameterizations based on chl $a$ that show persistent model biases. Thereafter, we seek to address this problem by proposing new approaches to estimate regional HVOC fluxes using airborne observations. We demonstrate two additional approaches for deriving HVOC flux estimates using airborne observations, and model output. We hope that the reviewer finds our article suitable for publication following these revisions.

L34-38 The regional enrichment ratios should be put in context here - there is no explanation as their relevance.

We no longer report enrichment ratios in the abstract. We do however, attempt to explain the role of $O_2$-HVOC enrichment ratios in inferring a biological flux of HVOCs. This passage now reads, "The first approach takes advantage of the robust relationships that were found between airborne observations of $O_2$ and $CHBr_3$, $CH_2Br_2$, and $CHClBr_2$; we use these linear regressions with $O_2$ and modeled $O_2$ distributions to infer a biological flux of HVOCs." L30-33.

L51- 52 "Indeed, HVOCs may be among the most important sources of inorganic bromine to the whole atmosphere ..... (Murphy et al.,in review)." This is not conventional wisdom and thus quite a bold statement. Are the authors confident that the Murphy et al paper will be published soon?

Murphy et al. (2019) has now been published and the citation has been revised. We have also moderated the language to reflect that this statement challenges conventional wisdom. This passage L50-54 now reads, "In the marine boundary layer and lower troposphere, sea salt is the main source of reactive bromine (Finlayson-Pitts 1982, Simpson et al. 2019). Yet HVOCs may also be a more important source of inorganic bromine to the whole atmosphere than previously thought, according to a recent study, which indicates that sea salt is scarce and insufficient to control the bromine budget in the middle and upper troposphere (Murphy et al., 2019)."

L61-64 The anthropogenic sources of CH3Br have changed over time and now are dominated by quarantine and pre-shipment (QPS) applications (not controlled by the Montreal Protocol). Please stick to the most recent information from WMO 2018 (and update the reference).

Both the information and citation on anthropogenic sources of $CH_3Br$ have been revised in L65-68: "$CH_3I$ is also formed through non-biological reactions in surface seawater, and $CH_3Br$ is emitted as a result of quarantine and pre-shipment activities, which are not regulated by the Montreal Protocol (e.g., Moore and Zafiriou; 1994, Engel and Rigby, 2018).

Elizabeth Asher 9/3/2019 11:46 AM

L119- 130 The last paragraph of the introduction would benefit from an introduction to the concept of enrichment ratios of HVOCs to O2, which feature prominently in the abstract.

We have revised this passage in L122-139 to read, "In Section 3.1 and 3.2, we report new airborne observations of $CHBr_3$, $CH_2Br_2$, $CH_3I$, $CHClBr_2$, $CHBrCl_2$, and $CH_3Br$ from high latitudes in the Southern Hemisphere, where data are scarce, and large-scale regional mixing ratio comparisons for HVOCs with the community earth system model (CESM) atmospheric component with chemistry (CAM-Chem). In section 3.4, we present two novel approaches to estimate regional fluxes of HVOCs for comparison with global climate models' parameterizations or climatologies. One approach uses correlations of HVOCs to marine, oxygen ($O_2$) of marine origin, as measured by deviations in the ratio of $O_2$ to nitrogen ($N_2$) ($\delta(O_2/N_2)$ see Sect. 2.1.2 and 3.1.2) to determine the importance of regional biological HVOC sources. The robust correlations of $CHBr_3$ and $CH_2Br_2$ with $\delta(O_2/N_2)$ are indicative of a strong biological source. Our first approach exploits the ratio of HVOCs to oxygen ($O_2$) determined from linear regressions (i.e. the enrichment ratio), and the ocean flux of $O_2$ from CESM's ocean component, to estimate the marine biogenic flux of several HVOCs. The second approach relies on observed HVOC mixing ratios, the Stochastic Time-Inverted Lagrangian Transport (STILT) particle dispersion model and geophysical datasets (see Sect. 2.3 and 3.3). We assess contributions from previously hypothesized regional sources for the Southern Ocean, and estimate HVOC fluxes based on regressions between upstream influences and observed mixing ratios and distributions of remotely sensed data."

L235-245 The fact that the polyhalogenated bromocarbons are likely co-emitted is not new – there are numerous papers that show this, and the discussion could elaborate on those a bit more. What is also missing from this paragraph is a discussion of macroalgal sources of these compounds, although this is presumably not relevant for the Antarctic.

We have expanded the discussion of previous findings of co-emitted polyhalogenated bromocarbons and cited several additional studies. This passage L390 - 401 now reads, "Previous studies have documented co-located source regions of $CHBr_3$ and $CH_2Br_2$ in the Southern Ocean (e.g. Hughes et al., 2009; Carpenter et al., 2000; Nightingale et al. 1995; Laturnus et al. 1996), and laboratory studies have demonstrated that phytoplankton and their associated bacteria cultures, including a cold water diatom isolated from coastal waters along the Antarctic Peninsula and common to the Southern Ocean, produce both $CHBr_3$ and $CH_2Br_2$ (Hughes et al., 2013; Tokarczyk and Moore 1994, Sturges et al., 1993). The non-linearity observed in ratios of these two gases at low $CHBr_3$ may reflect the different rates of their production or loss in seawater, or possibly, the influence of air masses from distant, more productive low-latitude source regions. Several studies have documented bacterially mediated loss of $CH_2Br_2$, but not

CHBr$_3$, and report distinct ratios of CH$_2$Br$_2$ to CHBr$_3$ in seawater during the growth and senescent phases of a phytoplankton bloom (e.g. Carpenter et al. 2009, Hughes et al 2013). "

L244-245 "For instance, Huges et al. (2013) also report distinct seawater slopes between CH2Br2 to CHBr3 , when chl a was increasing." It is not clear what is meant by this. Please rephrase

This statement has been rephrased on L398, "Several studies have documented bacterially mediated loss of CH$_2$Br$_2$, but not CHBr$_3$, and report distinct ratios of CH$_2$Br$_2$ to CHBr$_3$ in seawater during the growth and senescent phases of a phytoplankton bloom (e.g. Carpenter et al. 2009, Hughes et al 2013)."

L361- 366 "In both regions, the model under predicts CH3I above the MBL, which may indicate slower observed photochemical loss than the model predicts." Has this been found in other CAM-Chem studies – e.g. is it a general result? If not, could a different source emission distribution (i.e. more homogeneous source) explain these results?

We have revised the text to reflect that indeed this result has been found in other CAM-Chem studies, and that the observed difference at high latitudes in the SH at ~10 km altitude may be due to the zonal transport of air masses from lower latitudes, where differences in CH$_3$I in the UTLS have also been observed.  For instance, in Ordonez et al. (2012), Fig. 10 illustrates the consistent under prediction of the observed CH$_3$I mixing ratios, and these authors attribute this discrepancy to the strength of convective cells rapidly transporting air masses to the UTLS.  This section L494-499 now reads as follows: "In both regions, the model most likely under predicts CH$_3$I in the upper troposphere and lower stratosphere (UTLS), likely stemming from the poleward transport of lower latitude air masses, where CAM-Chem also exhibits a negative bias.  Mixing ratio comparisons with CAM-Chem over the tropics (see Ordonez et al. Figure 10) depict similar or larger discrepancies, and have been attributed to stronger than anticipated convective cells in the tropics."

L555-L560 onwards. There is no mention in Moore and Zarifou 1994 nor Richter and Wallace 2004 as far as I can see on the influence of iron availability – do the authors mean iodide availability?!

We have both fixed a typo and clarified the discussion on proposed non-biological chemical mechanisms for CH$_3$I production in the ocean, which include the radical recombination reaction proposed by Moore and Zarifou (1994), and the substitution reaction, requiring an oxidant such as iron III, proposed by Williams et al. (2007).  This passage L563-569 now reads, "This non-biological source, though not fully understood, requires light, a humic like substance at the surface ocean supplying a carbon source and methyl group, and reactive iodine (Moore and Zarifou 1994; Richter and Wallace 2004). Thus far, two chemical mechanisms have been proposed for the non-biological production of methyl iodide, one – a radical recombination of a methyl group and iodine involving UV photolysis (e.g. Moore and Zarifou 1994), and the second, a substitution reaction involving the reduction of an oxidant, such as iron III (e.g. Williams et al. 2007)."

L1036 – Note that units should be pmol m-2 hr-1 (not m2). Please state whether the values given for the observations are means or medians. It would be also be good to include their ranges.

We have corrected this typo on L1068.  The units on Table 1 now read "pmol m$^{-2}$ hr$^{-1}$."

Ln 82. atmopsheric Ln 213. "oppose" should be "opposed" Ln 213. "Huges" should be "Hughes" Ln 242 : "HOVCs" Ln 469. "Zafarou" should be "Zafariou" Ln 980. "includind" LN 1015. "fluxed"

L81, L253, L391, L518, L1171-  Typos have been corrected to read, "Atmospheric," "opposed," "HVOCs," "Zafiriou," and "fluxes."  Other typos previously listed have been deleted from the text.

Response to Reviewer #2

The paper presented by Asher et al., provides a valuable contribution to the understanding of the distribution and sources of halogenated organic substances from the Southern Ocean. The work publishes airborne observations of VHOC in an understudies region and applies new concepts for source determinations related to measurements of O2 and CO2 and geophysical datasets. It underlines current knowledge of the biological sources of CHBr3 and CH2Br2 by applying their ratio to oceanic oxygen emissions. CH3I appears to have a dominant biological source in the area of the Patagonian shelf, while closer to Antarctica a photochemical source appears to be dominant. The paper also compares the derived emissions of the novel concepts to the output of a global climate model. I agree that the presentation of data from several compounds and several campaigns is a difficult task. Also the loaded content of the paper: evaluation of model predictions, calculation of biogenic enrichment ratios, identification of regional sources, and novel means of parameterizing ocean fluxes, which was only summarized clearly in the conclusion section of the paper, makes the task of writing not easier. While the authors present their results and outcomes, which are not totally exciting and sometimes are also not very convincing due to poor correlations, their novel approaches and novel concepts are more exciting, but are poorly presented. They could do a much better job in explaining and presenting their concepts and the overall goal of the paper, which for me remains more a concept than a result paper. The results underline the novel approaches, as they do not contradict earlier studies and the novel approaches can be more useful and should be tested for and in future studies. The authors should think about a different setup of their paper, putting their concepts more into the focus, but clearly their approaches need be described more clearly and in more detail throughout the text. Also they authors should think about the title. Also technically the paper needs improvement, as abbreviations are sometimes not introduced, sometimes edits are not clearly overworked, which led to typos and grammar mistakes and also I wonder if it would be possible to make some sentences less bulky and loaded. Some figures are too small, some legends appear odd and there appear to be misunderstandings with some references . Overall I think the work behind the paper is very valuable and should be published in ACP, but the presentation of the work needs prior strong improvement.

We appreciate the reviewer's constructive criticism. We have refocused our paper on the approaches and concepts outlined here, rather than our results. We also argue that emissions of HVOCs over the Southern ocean are poorly known and seek to address this problem by proposing new approaches to estimate regional HVOC fluxes using airborne observations. We have sought to better outline the two novel approaches to estimate HVOC fluxes and explain why these approaches represent an important step forward in the field.  We have also done our best to improve the presentation by reorganizing the structure of the paper, simplified the language and corrected typos and grammar errors.  We hope that the reviewer finds our article suitable for publication following these revisions.

L22- 25 We also use CH3Br from the University of Miami Advanced Whole Air Sampler(AWAS) on ORCAS and from the UC Irvine Whole Air Sampler (WAS) on ATom-2. In connection with the first and the third sentence, this is a strange sentence. I think there is too much detail in the first two sentences about the instrumentation , which could be abandoned for the abstract and only explained later in the text.

We agree with the reviewer and have revised the text to include less detail on instrumentation.  This section on L22-25 now reads: "We present observations of $CHBr_3$, $CH_2Br_2$, $CH_3I$, $CHClBr_2$, $CHBrCl_2$, and $CH_3Br$ during the $O_2/N_2$ Ratio and $CO_2$ Airborne Southern Ocean (ORCAS) study and the 2nd Atmospheric Tomography mission (ATom-2), in January and February of 2016 and 2017."

L32-38 Based on these relationships What does this refer to?... it is unclear

We have done our best to clarify how the regressions of HVOC mixing ratios with upwind influences and $O_2$ are used to estimate basin-wide fluxes on L30-33: "… we demonstrate two novel approaches to estimate HVOC fluxes; the first approach takes advantage of the robust relationships that were found between airborne observations of $O_2$ and $CHBr_3$, $CH_2Br_2$, and $CHClBr_2$; we use these linear regressions with $O_2$ and modeled $O_2$ distributions to infer a biological flux of HVOCs."

L49-53 Indeed, HVOCs may be among the most important sources of inorganic bromine to the whole atmosphere, since recent evidence indicates that sea salt is scarce.. This is not true, as there is enough literature out to show how sea salt aerosol dominates the bromine in the lower troposphere. If the authors want to keep this sentence they have to provide more evidence, than just an upcoming paper. I suggest rewriting and specifying the statement to the known literature.

We have added a sentence to reflect that sea salt aerosol is critical to the bromine budget in the lower troposphere, and have moderated the language of the sentence regarding the contribution of bromocarbons to the middle and upper troposphere to reflect that this statement challenges conventional wisdom. This passage L50-54 now reads, "In the marine boundary layer and lower troposphere, sea salt is the main source of reactive bromine (Finlayson-Pitts 1982, Simpson et al. 2019). Yet HVOCs may also be a more important source of inorganic bromine to the whole atmosphere than previously thought, according to a recent study, which indicates that sea salt is scarce and insufficient to control the bromine budget in the middle and upper troposphere (Murphy et al., 2019)."

L75-80 There is an important observational paper missing which the authors need to relate to in the discussion of their results later on in the paper. It is: Regional sinks of bromoform in the Southern Ocean from 2013 from Mattsson et al. in GRL, where he shows the heterogeneity of the sources, which make the ocean a sink at times. Therefore also the next sentence needs to be revised: These studies indicate moderate ocean sources of CHBr3 and CH2Br2 at high latitudes in the Southern Hemisphere, and refer to Mattsson., possibly in line 79.

This passage has been revised on L79-84: "Mattsson et al. (2013) noted that the ocean also acts as a sink for HVOCs, when HVOC undersaturated surface waters equilibrate with air masses transported from source regions. The spatially heterogeneous ocean sources of $CHBr_3$ and $CH_2Br_2$ at high latitudes in the Southern Hemisphere are often underestimated in global atmospheric models (Hossaini et al., 2013; Ordoñez et al., 2012; Ziska et al., 2013)."

L94-103 Here you need to relate to Salawitch (2011?).. –most trace gases in tropospheric air enter the stratosphere in the tropics, move poleward and descend to the troposphere at middle and high latitudes. Salawitch claims, that the polar bromine can be influenced by large scale subsidence from the lower latitudes. . .

This passage has been rewritten to reflect this on L99-103, "As a result of limited vertical transport in these regions, however, air-sea fluxes lead to strong vertical gradients. Zonal transport from lower latitudes has a large impact on the vertical gradients of trace gas mixing ratios over polar regions (Salawitch 2010). Given their extended photochemical lifetimes at high latitudes (see Sect. 2.3 for a brief discussion), many HVOC distributions are particularly sensitive to zonal transport at altitude."

L105-106 Few constraints on HVOC mixing ratios or emissions based on airborne data exist at high latitudes in the Southern Hemisphere. What does this mean?

This sentence has been rewritten (L108-109) and is hopefully more clear: "Few airborne observations of HVOCs exist at high latitudes in the Southern Hemisphere."

L117 This is pmol. . . not nmol mol -1

The correction has been made on 119, "ACE-1 measurements of $CH_3I$ in the MBL indicate a strong ocean source between 40° S and 50° S in austral summer, with mixing ratios above 1.2 pmol below ~1 km (Blake et al., 1999)."

L131 – 136 Here you could (you need to do it somewhere) elaborate on the O2/N2 concept and why you chose to relate the HVOHCs to those.

We now discuss the concept and purpose of relating HVOCs to $O_2/N_2$ in L126-L134: "In section 3.4, we present two novel approaches to estimate regional fluxes of HVOCs for comparison with global climate models' parameterizations or climatologies. One approach uses correlations of HVOCs to marine, oxygen ($O_2$) of marine origin, as measured by deviations in the ratio of $O_2$ to nitrogen ($N_2$) ($\delta(O_2/N_2)$ see Sect. 2.1.2 and 3.1.2) to determine the importance of regional biological HVOC sources. The robust correlations of $CHBr_3$ and $CH_2Br_2$ with $\delta(O_2/N_2)$ are indicative of a strong biological source. Our first approach exploits the ratio of HVOCs to oxygen ($O_2$) determined from linear regressions (i.e. the enrichment ratio), and the ocean flux of $O_2$ from CESM's ocean component, to estimate the marine biogenic flux of several HVOCs."

L141-144 Please include the regions into Figure 1.

We now include the regions in Fig. 1.

L144-152 If you only refer to two flights from ATom-2, the sentence could be easier to read.

This sentence has been revised to read, "On Feb. 10 and 13, 2017 the sixth and seventh ATom-2 research flights passed over the eastern Pacific sector poleward of 60° S (defined here as Region 1) and over the Patagonian Shelf between 40° S and 55° S and between 70° W and 55° W (defined here as Region 2), respectively."

L204 Model is missing

Although this passage has bee revised, "Model" has been added when CESM is first introduced in the introduction on L126: "In Section 3.1 and 3.2, we report new airborne observations of $CHBr_3$, $CH_2Br_2$, $CH_3I$, $CHClBr_2$, $CHBrCl_2$, and $CH_3Br$ from high latitudes in the Southern Hemisphere, where data are scarce, and large-scale regional mixing ratio comparisons for HVOCs with the community earth system model (CESM) atmospheric component with chemistry (CAM-Chem)."

L207 You did not introduce CAM.

CAM is now referred to here as "CESM's atmosphere component. Please see above comment.

L1215 What is a broadening effect?

L217 We have specified "pressure broadening effect" on the CO2 and CH4 spectrum in cavity ring down instruments, which has been observed in several studies due to the influence of water vapor (e.g. Chen et al. 2013). This sentence now reads, "Dry-air mole fractions were calculated using empirical corrections to account for dilution and pressure broadening effects as determined in the laboratory before and after the campaign deployments, and in-flight calibrations were used to determine an offset correction for each flight."

L243-L252 We note that the non-linearity observed in ratios of these two gases at low CHBr3 levels likely reflects the differences in emissions during strong phytoplankton blooms, as oppose to other periods. The ratio may simply (and more likely) reflect other air masses from more distant source regions, which is reflected in a ratio which favors the longer lived compound (CH2Br2) over the shorter- lived compound (CHBr3) which is emitted in larger quantities in a biological source region (refer to Yokouchi, 20xx) but more rapidly degraded during transport.

Our analysis focuses on the bottom 2 km of the atmosphere, and as such largely reflects recent enhancements in HVOCs. Nevertheless, we have clarified this passage to reflect that contributions from zonal transport from low latitude regions cannot fully be ruled out, and have further expanded on the differences in CH2Br2 and CHBr3 production and loss rates in surface waters. This passage L395-405 now reads, "The non-linearity observed in ratios of these two gases at low $CHBr_3$ may reflect the different rates of their production or loss in seawater, or possibly, the influence of air masses from distant, more productive low-latitude source regions. Several studies have documented bacterially mediated loss of $CH_2Br_2$, but not $CHBr_3$, and report distinct ratios of $CH_2Br_2$ to $CHBr_3$ in seawater during the growth and senescent phases of a phytoplankton bloom (e.g. Carpenter et al. 2009, Hughes et al 2013). Although this analysis is restricted to the bottom 2 km of the atmosphere, zonal transport of air masses with lower ratios of $CH_2Br_2$ to $CHBr_3$ ratios, as have been observed in the MBL over productive, low-latitude regions, may also have influenced our observations (Yokouchi et al. 2005)."

L246-248 For instance, Huges et al. (2013) also report distinct seawater slopes between CH2Br2 to CHBr3 , when chl a was increasing. This is a weak sentence; can you give it more meaning?

This sentence has been rewritten on L398, "Several studies have documented bacterially mediated loss of $CH_2Br_2$, but not $CHBr_3$, and report distinct ratios of $CH_2Br_2$ to $CHBr_3$ in seawater during the growth and senescent phases of a phytoplankton bloom (e.g. Carpenter et al. 2009, Hughes et al 2013)."

L257-258 please explain the concept: What do you expect from the ratio of the HVOHCs and the marine oxygen.

L415-418 We have revised this passage, "We sought to test if the biologically mediated production of bromocarbons and oxygen result in similar atmospheric distributions. Conversely, we expected HVOC atmospheric distributions and $CO_2$ distributions to anticorrelate because $CO_2$ fixation in surface waters is proportional to the production of oxygen."

L288 238/241-242: Where did you get this equilibration times? Support them by reference or evidence. And also the air-sea fluxes of O2 and CHBr3 are not very similar. Revise. L444 This sentence now reads, "The bulk air-sea equilibration time for an excess of $CHBr_3$ and other HVOCs is less than two weeks, although the photochemical loss of HVOCs will alter their ratio over time (see Supplement for details on calculations of bulk sea air equilibration times)." The section in the supplement (L1177-1185) reads as follows: "To support the interpretation of our results, we calculate nominal equilibration times. For estimates of bulk sea air equilibration times for HVOCs, $O_2$, and $CO_2$, we assume a mixed layer depth of 30 m, a temperature of 0° C, a salinity of 35 PSU, and carbonate buffering according to eq. 8.3.10 in Sarmiento and Gruber (2006), and transfer velocities according to Nightingale et al., (2000). The Schmidt number (i.e. the ratio of the kinematic viscocity of a gas, divided by the molecular diffusivity) for $O_2$, $CO_2$ and $CH_3Br$ were calculated according to Wanninkof (2014), and the Schmidt numbers for $CHBr_3$ and $CH_3I$ were calculated according to Quack and Wallace (2003) and Moore and Groszko (1999), respectively. The results are provided in Sect. 3.1.2."

L291-300 This paragraph is a little back and forth between compounds and regions; it can be sorted for easier reading.

We have done our best to clarify this paragraph in L455-465: "Our observations suggest a biological source for $CHBr_3$ and $CH_2Br_2$ in both Region 1 and Region 2 (Fig. 4). Interestingly, the slope of the regression between $CHBr_3$ and $O_2$ appears distinct in Region 1 and Region 2, but between $CH_2Br_2$ is the same. Molar enrichment ratios are $0.20 \pm 0.01$, and $0.07 \pm 0.004$ pmol : mol for $CHBr_3$ and $CH_2Br_2$ to $O_2$ in Region 1, and $0.32 \pm 0.02$, and $0.07 \pm 0.004$ pmol : mol in Region 2. We observe a weaker relationship between $CH_3I$ and $CHClBr_2$ and $O_2$ in Region 1 (Fig. 4c, d), consistent with the existence of other, non-biological sources of $CH_3I$ in this region. Figure 4f illustrates a strong relationship between $CH_3I$ and $O_2$, as well as $CHClBr_2$ and $O_2$, in Region 2, however, which implies that the dominant sources of $CH_3I$ and $CHClBr_2$ emissions over the Patagonian Shelf are biological. The corresponding molar enrichment ratios of $CH_3I$ to $O_2$ and $CHClBr_2$ to $O_2$ in Region 2 are $0.38 \pm 0.03$ pmol : mol and $0.19 \pm 0.04$ pmol: mol, respectively."

263. This should have come earlier, when you start with the equilibria (238, 241). And do you also reference the atmospheric lifetimes?

L44 We now refer the reader to the supplement here for further reading on the calculation of equilibration times. Please see two responses up for details.

L336 – 337 FINN and MEGAN 2.1 products. I guess the abbreviations need to be explained a bit as well as the products

L247 "The model uses chemistry described by Tilmes et al. (2016), biomass burning and biogenic emissions from the Fire INventory of NCAR (FINN; Wiedinmyer et al. 2011) and MEGAN (Model of Emissions of Gasses and Aerosols from Nature) 2.1 products (Guenther et al., 2012) with additional tropospheric halogen chemistry described in Fernandez et al. (2014) and Saiz-Lopez et al. (2014), including ocean emissions of $CHBr_3$, $CH_2Br_2$, $CHBr_2Cl$, and $CHBrCl_2$, with parameterized emissions based on chlorophyll $a$ (chl $a$) concentrations and scaled by a factor of 2.5 over coastal regions, as opposed to open ocean regions (Ordoñez et al. 2012)."

L341 from this sentence it is not clear where the oceanic emissions are derived from. I guess its Ordonez, 2012? Done. Ordonez et al. (2012) has been cited. Please see above.

L343 Ordonez, 2012 does not include CH3Br. Revise

We respectfully disagree with the reviewer. Indeed, Ordonez does prescribe a lower boundary condition for $CH_3Br$ and show mixing ratio comparisons for this compound. There is not a biogenic flux prescribed for $CH_3Br$.

L393 GDAS has to be introduced.

L271 "STILT was run using 0.5° Global Data Assimilation System (GDAS) reanalysis winds to investigate the transport history of air sampled along the flight track (Stephens et al., 2018)."

416-418 We consider the wind direction error to evaluate the possible size of spatial errors in footprint location. There appears to be something wrong with the grammar? The sentence is not understandable. Given median wind speeds in this domain, this corresponds to a possible error of 260 km/day possible error. Here is also something wrong.

L287 We have revised this passage to read, "For wind speed, a small bias may be present, where we find a median difference between observations and reanalysis of 0.68 m/s, a 5% relative bias. The 1-sigma of the wind speed difference is 2.3 m/s, corresponding to a 19% 1-sigma uncertainty in wind speed. In its simplest approximation, the surface influence strength error is perfectly correlated with the wind speed error, and thus we take 19% as an approximation of the surface influence strength uncertainty. The uncertainty in surface influence location depends on the error in the wind direction. We find a 1-sigma error of 14 degrees in wind speed, which corresponds to a possible error of 260 km/day."

 OCI and GIOP have to be introduced. What does . . ..and its uncertainty . . .mean? how do you obtain a 0.25∘ x 0.25∘ gridded uncertainty in the detrital material absorption? It is also not clear from section 5.2.

OCI and GIOP are introduced, and we have done our best to clarify the meaning of GIOP absorption uncertainty in L350-362: "Due to persistent cloud cover over the Southern Ocean, which often precludes the retrieval of remotely sensed ocean color data, we used 8-day mean composite Aqua MODIS L3 distributions of chl $a$ from the Ocean Color Index (OCI) algorithm and absorption due to gelbstoff and detrital material at 443 nm from the Generalized Inherent Optical Properties (GIOP) model (NASA Goddard Space Flight Center, 2014). Absorption due to gelbstoff and detrital material at 443 nm is used as a proxy for colored dissolved organic matter (CDOM; https://oceancolor.gsfc.nasa.gov/atbd/giop/). CDOM is hypothesized to be an important source of carbon for the photochemical production of $CH_3I$ (Moore et al., 1994). The GIOP model also publishes an uncertainty in the absorption due to gelbstoff and detrital material at 443 nm. Raw 4 km x 4 km data were geometrically averaged, based on lognormal probability density functions, to a spatial resolution of 0.25° x 0.25° for use with gridded surface influences. We used the ratio of the 0.25° x 0.25° gridded uncertainty in the detrital material absorption to the absorption as the relative uncertainty for flux calculations (see Sect. 3.4.2)."

L477 and elsewhere – Is the new terminology geophysical influence function something different than the surface influence function? Or why do you change the wording? Its unclear.

We do not mean to confuse the reader with superfluous terminology: "geophysical influence function" has been replaced everywhere with "surface influence function."

to 404. Can you give an example for H and s. What is the potential geophysical source distribution s?

H is the surface influence based on a sample's back trajectories in the boundary layer (ppt $m^2$ s $pmol^{-1}$). An example of s would be the distribution of chl. a at the ocean surface ($\mu g\ m^{-3}$) or the distribution of fractional sea ice at the ocean surface, which is unitless.

412: here the potential source distributions is Hs1, Hs2. . .? And not s? Is HS1 the same as Hs1? 415-416: We used the standard deviation of the regression coefficients and the relative uncertainty in the source fields, added in quadrature, to estimate the uncertainty in these fluxes (see Fig. 7 and Sect. 5.2 for fractional uncertainties). 418: How did you calculate and do you report the relative uncertainty of the regression coefficients? There is no standard deviation of the regression coefficients in Fig 7 and sect 5.2 does not explain fractional uncertainties and no explanation is found about relative uncertainties in source fields. Or are you relating to surface influence strength uncertainty here. There needs to be more explanation about this added here.

L316-L333 Yes Hs1 is the same as HS2. This passage has been revised and two capitalization typos have been corrected to clarify the role of upstream influence functions and geophysical source distributions in these regressions with surface influence functions. Also an example of a geophysical source distribution s, was given, Chl. $a$, now L304. The relative uncertainty of regression coefficients for Figure 9 is reported, and used to calculate the flux shown in Figure 11 as described in Sect. 3.4.2. To clarify, in those regressions where a flux was not calculated based on the relationship (e.g. Fig 7-8), the uncertainty in the regression coefficients is not reported.

L501 why did you include. . . such as CH3I in Region 1? The second half sentence does not add information?

The phrase "such as CH3I in Region 1" has been deleted.

L500-501 Note, sea ice did not include land ice; however, we also found a negative correlation between upstream land ice influence and mixing ratios of HVOCs. Why do you add the sentence starting with however? How did you get the correlations when it is not included and does it help the interpretation of the results? It appears misleading and redundant.

This statement on L521 has been revised to read, "We found no positive relationships between upstream sea-ice influence and any measured HVOC Region 1 (Fig. 7)."

L506 We note that over-turned first year sea-ice, which can expose under-ice algae colonies to the air, likely still present a local source of CHBr3, CH2Br2, or other VOHCs to the MBL. How does this speculation relate to your study and how does it help your interpretations? It stands a bit loose currently.

The statements regarding land ice and overturned first-year ice have been deleted.

Sect. 4.2 What was the temporal resolution of the input data shortwave and detrital material- add in section 4.2.

The temporal resolution of the input shortwave radiation data is every six hours and detrital data is every eight days, as specified elsewhere on L351 (a) and L365 (b).

a) "Due to persistent cloud cover over the Southern Ocean, which often precludes the retrieval of remotely sensed ocean color data, we used 8-day mean composite Aqua MODIS L3 distributions of chl *a* from the Ocean Color Index (OCI) algorithm and absorption due to gelbstoff and detrital material at 443 nm from the Generalized Inherent Optical Properties (GIOP) model (NASA Goddard Space Flight Center, 2014)."

b) "The National Center for Environmental Prediction (NCEP) provides Final Global Data Assimilation System (GDAS/FNL) global data of downward shortwave radiation at the surface at 0.25 degree and 6-hour resolution (NCEP, 2015)."

L557-562 This section is wrong. There is no study (at least not the referenced ones) which proves a relationship between iron availability and methyl iodide. The authors have misinterpreted the cited studies. Please check and revise.

L565-569 A typo has been corrected and this passage has been revised and clarified. The role of iron is briefly explicitly discussed as a possible oxidant for one of two proposed abiotic $CH_3I$ reactions: "This non-biological source, though not fully understood, requires light, a humic like substance at the surface ocean supplying a carbon source and methyl group, and reactive iodine (Moore and Zarifou 1994; Richter and Wallace 2004). Thus far, two chemical mechanisms have been proposed for the non-biological production of methyl iodide, one – a radical recombination of a methyl group and iodine involving UV photolysis (e.g. Moore and Zarifou 1994), and the second, a substitution reaction involving the reduction of an oxidant, such as iron III (e.g. Williams et al. 2007)."

L564-565 citing the link between temperature and PAR to the solar radiation..this wording is strange..also add which temperature is needed. . .water . may be its easier to just write revealing the link to solar radiation ..or similar

L570 Done, this statement has been revised, "Several previous studies have correlated mixing ratios of $CH_3I$ to satellite retrievals of PAR and surface ocean temperature, revealing a link to solar radiation (e.g. Happell et al., 1996; Yokouchi et al., 2001)."

L424 please introduce TUV. This section appears to beat the wrong place. I would expect this earlier in the description of the model, e.g. in 4.1., where you also talk about uncertainties due to meteorology.

TUV is now introduced. Note, this section L419-429 has been moved up as suggested to the end of Sect. 2.3 L295-306: "Finally, we note that photochemical loss during transport is not accounted for in this analysis. Low OH mixing ratios, cold temperatures, and lower photolysis rates due to angled sunlight at high latitudes lead to longer than average HVOC lifetimes. For instance, assuming an average diurnal OH concentration of 0.03 pptv, and average photochemical loss according to the Tropospheric Ultraviolet and Visible (TUV) radiation model and the Mainz Spectral data site (http://satellite.mpic.de/spectral_atlas) for Jan. 29 under clear sky conditions at 60° S, $CHBr_3$ has a lifetime of 30 days, $CH_2Br_2$ has a lifetime of 270 days, $CH_3I$ has a lifetime of 7 days, and $CHClBr_2$ has a lifetime of 63 days. As such, the photochemical lifetimes of these gases are greater than or equal to the time of our back-trajectory analysis. Moreover, OH concentrations in this region have large uncertainties, the inclusion of which would lead to more, not less, uncertainty in surface influence based regression coefficients and estimated fluxes."

L571-576.

as explained earlier the concept needs to be introduced more clearly earlier..e.g. why do you not take the VOHC directly but apply their relationship to oxygen?

Our goal was not to suggest the "correct" regional flux of HVOCs based on data from two austral summers (and relatively few measurements from the Atom-2 campaign in 2017), but to demonstrate that airborne data can be use to develop other empirically based parameterizations, which could work better. We argue that despite the its inherent uncertainties in the parameterization of biogenic HVOC fluxes based on $O_2$, the current CAM-Chem scheme based on chl. *a* leads to biases that exceed 50-100% for these compounds. Moreover, the uncertainties in remotely sensed chl.*a* are rarely considered in such parameterizations.

How can a model and an observation based flux-estimate be wrong by around 50%? And why do you think that a simple down scaling of the calculated oxygen fluxes leads to a robust flux estimate for VOHC, respectively why is this better, than taking just the VOHC fluxes? Can you explain this concept in the text please? Also it is unclear why you calculate all your influence functions to the VOHC mixing ratios directly, and not to their relation to oxygen and why for the flux calculation this now appears better?

$O_2$ and $CO_2$ fluxes are not well constrained at high latitudes in the southern hemispheres. In fact, the ORCAS campaign sought primarily to address this problem. Please see Stephens et al. (2018) for details. Although we agree with the reviewer that the simple downscaling is crude, this large discrepancy between observations and model or climatological mean values is due to inter-annual variability. The uncertainty discussed in L546-548 is meant to account for errors in the spatial variability in the fluxes, and does not include the mean absolute difference that is adjusted for in downscaling.

L632-634 I strongly believe that the calculation of the regression surface influence functions need to be shown in the text not in the legend of figure 9. Regression coefficients from the MLR with surface influence functions are now shown in here on L641-644 not in the legend of Fig. 9, "We used a multiple linear regression (± 1standard deviations; Equation 2), where Hs1 and Hs2 are the surface influence functions of downward shortwave radiation and detrital absorption, respectively, with an intercept b = 0.19 ± 0.01, and influence coefficients $a_1$ = 3.7E-5 ± 1.3E-5, $a_2$ = 3.5± 0.74, and an interaction term with the coefficient -5.2E-4 ± 1.5E-4 (c)."

Table 1. You need to indicate in table1 ,which method you used in "This study" to derive the reported flux, as there are several methods here.

The approaches ($O_2$ vs. MLR using surface influence functions) has been clarified here.

L678-680 this also appears true for CHBr3, CHClBr2 in region 1 ..and for the entire troposphere for CHBrCl2

We have rewritten this passage. L692-704: "Our flux estimates based on the relationship of HVOC mixing ratios to other airborne observations and remotely sensed parameters compared relatively well with those derived from global models and ship-based studies (Table 1). Our emission estimates of $CHBr_3$, $CH_2Br_2$, and $CHClBr_2$ are significantly higher than CAM-Chem's globally prescribed emissions in Region 1, where HVOC mixing ratios are under predicted (Table 1; Fig. 5). Similarly, our estimate of $CHClBr_2$ emissions is also significantly higher than CAM-Chem's in Region 2, where $CHClBr_2$ mixing ratios remained under predicted. Nevertheless, our emission estimates of $CHBr_3$, $CH_2Br_2$, $CH_3I$, are lower than most prior estimates based on either other models or localized studies using seawater-side measurements from the Antarctic polar region in summer. In the case of $CH_3I$, our estimated emissions suggest that the prescribed emissions in CAM-Chem may be too high in Region 1 and Region 2. Our parameterizations of the $CH_3I$ flux could be used to explore inter-annual variability in emissions, which is not captured by the Bell et al. (2002) $CH_3I$ climatology currently employed in CAM-Chem."

L660-664 although they were significantly higher than CAM-Chem's prescribed emissions in Region 1, where VOHC mixing ratios are under predicted (Table 1; Fig. 5). Can you please add the comparison to CAM- Chem at the beginning .It would be better structured if you don't jump between comparisons.

We have clarified these two passages. Please see above.

L675 – 684 parameterizations..these are different ..and you need to add which compounds you are referring to in this sentence. Here it would be good to extend on the methods and why they are appear so usefull and how you would extend them to other species.

L705-711 We have done our best to clarify this passage: "To extend these relationships to year-round and global parameterizations for use in global climate models, they must be studied using airborne observations in other seasons and regions. These approaches may help parameterize emissions of new species that can be correlated with surface influence functions or the biological production of oxygen or may improve existing emissions, where persistent biases exist. Finally, future airborne observations of HVOCs have the potential to further improve our understanding of air-sea flux rates and their drivers for these chemically and climatically important gases over the Southern Ocean."

Figure 2. Are the data of the campaigns merged? Detection limits need to be added. The label of CH3I is odd.

Yes, the data are merged. Detection limits have been added to the legend.

Figure 3. Please specify one name for the campaigns and keep it. Here in one figure the authors switch between Atom-2, Atom and Atom. Line 937 to 939 in the legend : This sentence does not make sense.

All mentions of ATom are now listed as ATom-2.

Figure 4. There appears to be an old legend as d, g and h are missing as well as CHClBr2. The applied regressions appear to be the same , thus it would be good to elaborate in the text about the method to reduce the legend, e.g. what means. using variables scaled to their range? In the legend? Also here only regressions above 0.2 are shown .

The legend has been revised, and a statement has been added to say that only regressions with $r^2 > 0.2$ are shown.

Figure 5,6. Switching between CESM in the figure and CAM-Chem1.2 does not help clarity. . . .multiplied by the percentage of data below detection.. .. was it used for calculating the mean? ..rephrase for clarity.

CESM in all the figure axes has been relabeled CAM-Chem. The sentence regarding data below the DL has been revised to read, "Again, the binned mean includes measurements below the detection limit (DL), which for this calculation are assigned a value equal to the DL multiplied by the percentage of data below detection."

Figure 7,8. Talking of statistical significance with r2 « 0.2 and looking at the plots with scattered values and no surface influence, is a bold exaggeration. And the p-values can be abandoned from the figure and just the threshold mentioned, as they do not help the statistics.

P-values listed on the plots have been replaced with p-value thresholds (e.g. $p < 0.001$).

Figure 9. The labeling of the figures is too small, the p- value redundant and the legend for figure c.) too intricate. I strongly believe that the calculation of the regression surface influence functions need to be shown in the text not in the legend of figure 9.

The size of figure labels is larger. The calculation is now shown in the text as discussed above.

Figure 10. The figures and labeling of a to c are too small. ( I suggest single plots, resolution as Figure 1?. It must be pmol m-2 hr-1 also in the legend. Also clarify that these are model results. How do the mentioned CESM (CAM- Chem 1.2) O2 fluxes relate to the figure? And is this also 2016?

The labeling is now larger, and as now stated in the figure legend, represents the year 2016. CESM ocean component $O_2$ fluxes (not shown here) were multiplied by the regression coefficients shown in Fig. 4 to infer a biological flux of HVOCs, as explained in Sect. 5.1.

Figure 11. fluxes not fluxed

Done, and now reads, "fluxes."

Response to Reviewer #3

Anonymous Referee #3 The manuscript of Asher et al. describes airborne observations of halogenated volatile organic compounds over the Southern Ocean and improved emission flux estimates, based on modeling studies and correlative O2 observations. This is an important and interesting study that should be published in Atmos. Chem. Phys. after consideration of the following points. The authors should consider improving the presentation by first presenting their data and methods and then discussing the results. This study contains important new methods and approaches compared to previous studies but the presentation is not always clear. As an example, a key result is the presentation of "regional enrichment ratios" for HVOCs, but it did not become sufficiently clear to me, how they are defined and how they were calculated.

We appreciate the reviewer's comments and suggestions. We have done our best to reorganize the paper accordingly, by first discussing our methods and data sources and discussing our results second. We have paid particular attention to clarifying the discussion of regional enrichment ratios for HVOCs in the abstract as well as in sections 3.12 and 3.31.

Specific Comments:

Specific comments: L32-34: in the same sentence "halogenated hydrocarbon" and "halogenated volatile organic compounds (HVOCs)" are used. If the two mean the same, use only one name. If there is a distinction, please define.

Indeed- these are the same. The wording has been revised. Only the term "halogenated volatile organic compounds (HVOCs)" is now used.

L47-49: Is there a particular logic for the order of the citations given? They are neither sorted according to year, nor alphabetically.

This has been corrected and special attention has been paid to the order of citations throughout the paper.

L50: "recent evidence indicates that sea salt is scarce and insufficient": this is a strong statement that should be backed up with more than a manuscript in review.

We appreciate the reviewer's comment. Although this study is now published, this statement has been amended to better reflect current understanding on L50 – 54: "In the marine boundary layer and lower troposphere, sea salt is the main source of reactive bromine. Yet HVOCs may also be a more important source of inorganic bromine to the whole atmosphere than previously thought, according to a recent study, which indicates that sea salt is scarce and insufficient to control the bromine budget in the middle and upper troposphere (Murphy et al., 2019)."

L66: You may cite Abrahamsson et al. (2018) already at this stage.

Done. L670: "Over the Southern Ocean specifically, hypothesized sources of HVOCs include: coastal macroalgae, phytoplankton, sea ice algae, and photochemical or dust stimulated non-biological production at the sea surface (e.g., Abrahamsson et al. 2018, Manley and Dastoor 1998; Moore and Zafiriou 1994; Moore et al., 1996; Richter and Wallace 2004; Williams et al., 2007; Tokarczyk and Moore 1994; Sturges et al., 1992)."

L96: The point "support quantitative air-sea flux estimates" is less obvious than the other points so a reference may be helpful here.

Thank you, we have revised this sentence on L106 to read, "Aircraft observations can rapidly map basin-wide vertical distributions, support quantitative flux estimates, and provide spatial constraints to atmospheric models (e.g. Xiang et al. 2010x; Stephens et al 2018; Wofsy et al. 2011)."

L211: "We note that the non-linearity observed in ratios of these two gases at low CHBr3 levels likely reflects the differences in emissions during strong phytoplankton blooms, as oppose to other periods." Could not the different lifetimes also effect this?

L395-409 Thank you, this passage has been amended to reflect this possibility, and we have done our best to clarify the wording: "The non-linearity observed in ratios of these two gases at low $CHBr_3$ may reflect the different rates of their production or loss in seawater, or possibly, the influence of air masses from distant, more productive low-latitude source regions. Several studies have documented bacterially mediated loss of $CH_2Br_2$, but not $CHBr_3$, and report distinct ratios of $CH_2Br_2$ to $CHBr_3$ in seawater during the growth and senescent phases of a phytoplankton bloom (e.g. Carpenter et al. 2009, Hughes et al 2013). Although this analysis is restricted to the bottom 2 km of the atmosphere, zonal transport of air masses with lower ratios of $CH_2Br_2$ to $CHBr_3$ ratios, as have been observed in the MBL over productive, low-latitude regions, may also have influenced our observations (Yokouchi et al. 2005)."

Fig. 3. Units missing for the axes

This has been corrected- thank you for brining it to our attention.

Fig. 4. Why are some units given as nmol/mol and others as ppt ?

This too has now been corrected, the axes all read ppt. Again, thank you for brining this to our attention.

L222: Sorry, but I don't know what a type II major axis regression is. A few more words may help.

L426-L431 We have added a short passage to clarify the meaning and utility of the type II major axis regression in this analysis: "We used a type II major axis regression model (bivariate) to balance the influences of uncorrelated processes and measurement uncertainty in HVOCs (on the y-axis) and uncorrelated processes and measurement uncertainty in $O_2$ and $CO_2$ (on the x-axis) on the regression slope (Ayers et al. 2001; Glover et al., 2011). As noted by previous studies, simple least squares linear regressions fail to account for uncertainties in predictor variables (e.g. Cantrell et al. 2008)."

L250: Please explain how the molar enrichment ratios are defined and/or calculated. This seems to be critical, but not well explained. Is this just the slope of the regression between CHBr3 (or CH2Br2) and O2?

Yes, the molar enrichment ratios are equivalent to the slope of the regression, although the units of $O_2$ must be converted from $O_2/N_2$ (per meg) to equivalent ppm (multiplying O2/N2 by the $XO_2$, in dry air = 0.2093).

L351: "In its simplest approximation, the wind speed error will correlate with surface influence error" I understand that this is in general may be a reasonable assumption, but it is not obvious to me why the error in the influence function (in ppt m2 s pmol-1) should be proportional to the error in wind speed. More justification of this argument would be needed here.

As explained in Xiang et al. 2010, now cited here, the STILT model error (E) represents a combination of source and model transport error. Although model transport error is difficult to quantify precisely, it is influenced first and foremost by differences in simulated and actual wind speed, wind direction, and boundary layer height. This passage L280-294 now reads, "Uncertainty in the surface influence value is strongly influenced by the accuracy of the underlying meteorological transport, as discussed in Xiang et al. (2010). We evaluated the GDAS reanalysis winds by comparing model winds interpolated in space and averaged between corresponding time points and pressure levels to match aircraft observations. By evaluating observed winds compared with modeled winds along the flight tracks we can estimate uncertainty in the surface influence values. We consider the observation-model differences in both wind speed and direction to approximate errors in surface influence strength and location. For wind speed, a small bias may be present, where we find a median difference between observations and reanalysis of 0.68 m/s, a 5% relative bias. The 1-sigma of the wind speed difference is 2.3 m/s, corresponding to a 19% 1-sigma uncertainty in wind speed. In its simplest approximation, the surface influence strength error is perfectly correlated with the wind speed error, and thus we take 19% as an approximation of the surface influence strength uncertainty. The uncertainty in surface influence location depends on the error in the wind direction. We find a 1-sigma error of 14 degrees in wind speed, which corresponds to a possible error of 260 km/day."

L389: PAR: please spell out (as far as I can see first defined in L476)

L369 Thank you. This is now done.

L431: "We note that over-turned first year sea-ice, which can expose under-ice algae colonies to the air, likely still present a local source of CHBr3, CH2Br2, or other HVOCs to the MBL." What is this statement based on?

As it is irrelevant to the main objective of the paper, this statement has been removed.

L499: Reference to Fig.9 in L499 was not clear to me. Was really Fig.9 meant here?

Fig. 10 is now referenced here, "For $CHBr_3$, $CH_2Br_2$, and $CHClBr_2$ we construct ocean emission inventories for January and February using a scaled version of gridded modeled air-sea $O_2$ fluxes and the slopes (i.e. molar ratios) of linear correlations between $\delta(O_2/N_2)$ and HVOC mixing ratios (Fig. 10)."

Fig. 9c: Caption not very clear, would be helpful if the description in the caption can be improved.

The wording of this caption has been rewritten. As now discussed elsewhere in the text (Sect. 2.3.1) the surface influence function (e.g. $HS_1$) is the product of the surface influence and a relevant surface source field.

5.2 Why are STILT based emission estimates presented only for CH3I? Why is it not possible to perform this for other HVOCs as well?

Indeed, it is possible to estimate STILT emissions for other gases such as CHBr3 and CH2Br2. At present, we have not done this, as the correlations with STILT surface influence functions were less strong than those with $O_2/N_2$, as now stated in the text L653-656.

Figure S4: "Consecutive samples in and out of dips into the MBL": Sorry, I don't really understand what is meant here, please re-word.

This has been reworded as requested to read, "Consecutive TOGA VOC sample locations, their back-trajectories and surface influences in the lower troposphere on two different flights (a-c; Jan. 21,2016, and d-f; Jan. 30, 2016)."

Technical corrections: L134: "low attitude" -> "low altitude"

Done.

L183: citation should be part of the sentence

Done.

[revised manuscript text omitted]

Elizabeth Asher 7/8/2019 5:53 PM

Elizabeth Asher 9/3/2019 11:16 AM

Elizabeth Asher 7/8/2019 5:53 PM

Elizabeth Asher 7/8/2019 5:53 PM

Elizabeth Asher 7/8/2019 5:54 PM

Elizabeth Asher 7/7/2019 12:02 PM

Elizabeth Asher 9/3/2019 11:16 AM

Elizabeth Asher 7/7/2019 12:10 PM

Elizabeth Asher 7/7/2019 12:10 PM

Elizabeth Asher 7/7/2019 12:29 PM

Elizabeth Asher 7/7/2019 12:30 PM

Elizabeth Asher 9/3/2019 11:17 AM

Elizabeth Asher 7/7/2019 12:30 PM

Elizabeth Asher 7/7/2019 12:31 PM

Elizabeth Asher 7/7/2019 12:31 PM

Elizabeth Asher 7/7/2019 12:31 PM

Elizabeth Asher 9/3/2019 11:17 AM

Elizabeth Asher 7/7/2019 12:33 PM

[revised manuscript text omitted]

**Supplementary Text**

Sea air exchange calculations

To support the interpretation of our results, we calculate nominal equilibration times. For estimates of bulk sea air equilibration times for halogenated VOCs, $O_2$, and $CO_2$, we assume a mixed layer depth of 30 m, a temperature of 0° C, a salinity of 35 PSU, and carbonate buffering according to eq. 8.3.10 in Sarmiento and Gruber (2006), and transfer velocities according to Nightingale et al., (2000). The Schmidt number (i.e. the ratio of the kinematic viscocity of a gas, divided by the molecular diffusivity) for $O_2$, $CO_2$ and $CH_3Br$ were calculated according to Wanninkof (2014), and the Schmidt numbers for $CHBr_3$ and $CH_3I$ were calculated according to Quack and Wallace (2003) and Moore and Groszko (1999), respectively. The results are provided in Sect. 3.1.2.

Comparisons of TOGA, WAS and PFP

Despite overall good agreement between co-located inflight AWAS, WAS, and PFP samples and TOGA measurements, we observed notable discrepancies in several cases (e.g. Fig. S1b; Fig. S2a-b). On ORCAS, we observed a non-linear relationship between inflight TOGA measurements and co-located AWAS samples of $CH_3I$ (Fig. S1b), driven by a few samples with high mixing ratios. Close inspection of upwind and downwind flights over Region 2 with the campaign's high mixing ratios of $CH_3I$ indicated that TOGA measurements were consistent with a modest flux of $CH_3I$ from the ocean to the atmosphere. On ATom-2, TOGA measurements agreed better with co-located PFP samples than with co-located WAS samples; and differences on the sixth and seventh research flights (i.e. the data used here) were relatively small. Nevertheless these differences motivated an instrument inter-comparison following the ATom campaign between these instruments. Thus far, results of this inter-comparison show that TOGA and PFP measurements differ by < 25%.

Supplementary Tables

Table S1. The TOGA-PFP instrument comparison was done by sampling a 50L SS pontoon, created at NCAR from a humidified dilution of the TOGA ATom standard. Data were analyzed and reported by Rebecca Hornbrook (NCAR, TOGA) and Steve Montzka (NOAA, PFP).

| Pontoon Inter-comparison | Concentration (dilution-based calc.) | TOGA (10/12/2018) | PFP (10/24/2018) |
|---|---|---|---|
| $CHBr_3$ | 34 | 21.0 ± 0.1 | 26.6 ± 0.8 |
| $CHClBr_2$ | 26 | 19.9 ± 1.0 | 22.9 ±0.1 |
| $CH_2Br_2$ | 52 | 47.7 ± 0.2 | 51.7 ± 2.0 |

**Supplementary Figures**

**Figure S1.** Comparison between AWAS samples and TOGA measurements during ORCAS below 10 km, when these two shared over half their sampling period  Points are colored by altitude. Dashed lines represent ± 10% of the 1:1 line.  Sample points below the DL are not included in this quantitative comparison.

[Figure]

**Figure S2.** Comparison between WAS, PFP and TOGA measurements during ATom-2 below 10
km, when these instruments shared over half their sampling period. WAS measurements are
shown in larger circles, PFP measurements in smaller circles, and measurements from the
research flights six and seven used in this analysis are shown in color, while measurements on
other research flights in ATom-2 are shown in gray.  Dashed lines represent ± 10% of the 1:1
line.  Sample points below the DL are not shown.

[Figure]

**Figure S3.** Mixing ratios of $CHBr_3$, $CH_2Br_2$ and $CH_3I$ vs. $CO_2$ in Region 1 (a-c) and Region 2 (d-
f). Type II major axis regression model (bivariate least squares regression) fits are shown for
combined ORCAS and ATom-2 data, using data below 7 km for $CHBr_3$, $CH_2Br_2$, and below 2
km for $CH_3I$.

[Figure]

**Figure S4.** Two sets of three consecutive TOGA VOC sample locations, their back-trajectories
and surface influences in the lower troposphere on two different flights (a-c; Jan. 21,2016, and d-
f; Jan. 30, 2016).  For illustrative purposes, sampling locations are denoted by a black circle, 24-
hour back trajectories are shown in red, and surface influences are shown with black squares in
each subpanel, overlying weekly composites of remotely sensed chl *a*.  Surface influence is
multiplied by the underlying chl *a* (or other) surface field and averaged for each sample to yield
a surface influence function.